# Rhythmic light flicker rescues hippocampal low gamma and protects ischemic neurons by enhancing presynaptic plasticity

Lifeng Zheng [1,4], Mei Yu[1,4], Rui Lin[1], Yunxuan Wang[1], Zhan Zhuo[1], Ning Cheng[2], Mengzhen Wang[1], Yongqiang Tang[3], Liping Wang[3] & Sheng-Tao Hou [1✉]

The complex relationship between specific hippocampal oscillation frequency deficit and cognitive dysfunction in the ischemic brain is unclear. Here, using a mouse two-vessel occlusion (2VO) cerebral ischemia model, we show that visual stimulation with a 40 Hz light flicker drove hippocampal CA1 slow gamma and restored 2VO-induced reduction in CA1 slow gamma power and theta-low gamma phase-amplitude coupling, but not those of the high gamma. Low gamma frequency lights at 30 Hz, 40 Hz, and 50 Hz, but not 10 Hz, 80 Hz, and arrhythmic frequency light, were protective against degenerating CA1 neurons after 2VO, demonstrating the importance of slow gamma in cognitive functions after cerebral ischemia. Mechanistically, 40 Hz light flicker enhanced RGS12-regulated CA3-CA1 presynaptic N-type calcium channel-dependent short-term synaptic plasticity and associated postsynaptic long term potentiation (LTP) after 2VO. These results support a causal relationship between CA1 slow gamma and cognitive dysfunctions in the ischemic brain.

[1] Brain Research Centre and Department of Biology, Southern University of Science and Technology, 1088 Xueyuan Blvd, Nanshan District, Shenzhen 518055 Guangdong Province, China. [2] The Shenzhen Second People's Hospital and the First Affiliated Hospital of Shenzhen University Health Science Center, Shenzhen 518035, China. [3] CAS Center for Excellence in Brain Science, Shenzhen Institutes of Advanced Technology, Chinese Academy of Sciences, Shenzhen 518055, China. [4] These authors contributed equally: Lifeng Zheng, Mei Yu. ✉email: hou.st@sustech.edu.cn

Changes in the oscillatory amplitude, frequency, and task-related brain rhythms are linked with human brain diseases, such as stroke[1–5]. A reduction in the 30–50 Hz low gamma oscillations in stroke patients correlates with reduced muscle strength and dynamics[5]. The hippocampal formation is sensitive to hypoxia-ischemia. However, the relationship between cerebral ischemia-induced cognitive dysfunction and the dynamic changes in hippocampal oscillations remains not clear.

Hippocampal gamma is associated with working memory and learning[1,6,7]. The perturbation of gamma underlies cognitive dysfunctions in neurological disorders[2,3,6]. Reduction in hippocampal local field potential (LFPs) cross-frequency coupling (CFC) between theta and gamma is associated with impaired long-term potentiation (LTP) in the anesthetized rat with transient global cerebral ischemia[8]. A long-lasting decrease in the low gamma, but not theta or high gamma, has also been observed in an anesthetized mouse model of unilateral hippocampal ischemia[9]. However, the role of hippocampal oscillation in conscious and behaving animals recovering from cerebral ischemia is not known.

Activity-dependent neural oscillations can be non-invasively and externally modulated by a 40 Hz rhythmic light flicker. Entrainment with visual cortex gamma oscillation attenuated cognitive dysfunctions in mouse models of Alzheimer's disease[9–12]. These exciting findings suggest that driving brain oscillation can restore cognitive dysfunctions. The present study aimed to determine whether modulation of hippocampal gamma could alleviate cerebral ischemia-induced cognitive dysfunctions.

Deficits of CA3-CA1 excitatory synaptic LTP, which is important for learning and memory[13–16], contribute to cognitive impairments in cerebral ischemia[17,18]. Transient and bilateral occlusion of the common carotid arteries (2VO) in mice selectively damages hippocampal CA1 pyramidal neurons when the occlusion time is empirically titrated[19], making 2VO a suitable system to interrogate the dynamic relationships between CA3-CA1 neural oscillation and cognitive dysfunctions following cerebral ischemia[8,18]. Here, we show rhythmic 30–50 Hz light flicker can restore hippocampal CA1 low gamma oscillation and protect ischemic CA1 neurons through a mechanism requiring RGS12-regulated N-type CaV2.2 voltage-gated calcium channels (N-VGCC)-dependent enhancement of synaptic plasticity.

## Results

**40 Hz light flicker entrained with hippocampal CA1 low gamma.** To demonstrate the effects of different frequencies of light flicker on CA1 oscillations, LFPs were recorded in mice implanted with a four-electrode array into the CA1 stratum pyramidale layer (Fig. 1a), followed by treatment with 10 Hz, 30 Hz, 40 Hz, 50 Hz, 80 Hz, arrhythmic frequency, or without light flicker (occluded LED) light flicker (Fig. 1b–d and Supplementary Fig. 1j–m). Because mouse movement speed affects the linearity of theta waves and potentially presents theta harmonics[20,21], three similar low-speed movement states, i.e., approach/exploration of a novel object and low-speed free walking, were selected for the LFP study. The speeds/velocities were at 2~3 cm/s forward, as determined by footfall patterns (Supplementary Fig. 1n)[22]. Oscillation entrainment occurred in CA1 LFP in response to light flickers of 40 Hz (Fig. 1c), 10 Hz, 30 Hz, 50 Hz and 80 Hz (Supplementary Fig. 1j – m), but not the occluded LED or arrhythmic light (Fig. 1b–d). The power spectral density (PSD) of CA1 low gamma, but not theta and high gamma, oscillation increased significantly in response to 40 Hz light flicker treatment compared with the controls (1ANOVA, $F_{(2, 15)} = 4.512$, $P = 0.0292$; LSD's post hoc test, $P_{\text{Occluded LED vs. 40 Hz LED}} = 0.0209$, $P_{\text{40 Hz LED vs. Random LED}} = 0.0192$, Fig. 1f), confirming previous studies[10,12,23,24].

**40 Hz light flicker restored CA1 low gamma reduction in 2VO.** 40 Hz light flicker restores 2VO-affected CA1 LFP. Mice received light flicker treatment 2 h after 2VO twice per day, 12 h apart, for up to 14 days (Supplementary Fig. 1a–d). 2VO group of mice CA1 LFP power (Fig. 1h) was significantly reduced in the 30–50 Hz low gamma range (2ANOVA, $F_{(1, 22)} = 7.628$, $P = 0.0114$; LSD's post hoc test, $P_{\text{Sham vs. 2VO}} = 0.0434$, Fig. 1j), but not 4–12 Hz theta (2ANOVA, $F_{(1,22)} = 0.04573$, $P = 0.8326$; LSD's post hoc test, $P_{\text{Sham vs. 2VO}} = 0.8454$, Fig. 1i) and 80–120 Hz high gamma (2ANOVA, $F_{(1,22)} = 0.7381$, $P = 0.3995$; LSD's post hoc test, $P_{\text{Sham vs. 2VO}} = 0.5716$, Fig. 1k). The 2VO mice also displayed decreased phase-amplitude coupling (PAC) between theta phase and slow gamma amplitude compared with the Sham group (2ANOVA, $F_{(1, 8)} = 5.441$, $P = 0.0480$; Tukey's post hoc test, $P_{\text{Sham vs. 2VO}} = 0.0006$, Fig. 1l). Theta-high gamma PAC was not altered in 2VO mice (2ANOVA, $F_{(1,8)} = 0.07855$, $P = 0.7864$; Tukey's post hoc test, $P_{\text{Sham vs. 2VO}} = 0.9370$, Fig. 1m). Importantly, 40 Hz light flicker treatment significantly enhanced 2VO group CA1 low gamma power (2ANOVA with LSD's post hoc test, $P_{\text{2VO vs. 2VO+LED}} = 0.0001$, Fig. 1j), and restored theta-low gamma PAC in the 2VO + LED group (2ANOVA with Tukey's post hoc test, $P_{\text{2VO vs. 2VO+LED}} = 0.0021$, Fig. 1l), indicating a specific reduction of hippocampal CA1 low gamma in cerebral ischemia.

Novel object exploration elicits a distinct CA1 40 Hz low gamma oscillation, which are related most specifically to object-location associative memory encoding rather than overt behaviors[25,26]. CA1 LFPs were therefore recorded in four groups of mice performing an approach/exploration of novel object task (Fig. 1n). An example of a moving window spectrogram of a mouse time-locked as a mouse initiated an approach (<0 s) and exploration (>0 s) of a novel object was shown in Fig. 1n. A significant reduction in the 30–50 Hz low gamma range and the 40 Hz specific gamma occurred in the PSD ratio between Exploration/Approach compared between Sham vs. 2VO group (For low gamma: RM-2ANOVA, $F_{(1, 10)} = 70.82$, $P < 0.0001$; Sidak's post hoc test, $P_{3d} < 0.0001$, $P_{7d} < 0.0001$, $P_{14d} = 0.0006$, Fig. 1p. For 40 Hz gamma: RM-2ANOVA, $F_{(1, 10)} = 144.60$, $P < 0.0001$; Sidak's post hoc test, $P_{3d} < 0.0001$, $P_{7d} < 0.0001$, $P_{14d} < 0.0001$, Fig. 1o). No change occurred in the 4–12 Hz theta (RM-2ANOVA, $F_{(1, 10)} = 0.637$, $P = 0.4433$; Sidak's post hoc test, $P_{3d} = 0.7603$, $P_{7d} = 0.9061$, $P_{14d} = 0.9959$, Fig. 1q) and the 80–120 Hz high gamma (RM-2ANOVA, $F_{(1, 10)} = 0.4482$, $P = 0.5184$; Sidak's post hoc test, $P_{3d} = 0.7685$, $P_{7d} = 0.9998$, $P_{14d} = 0.9473$, Fig. 1r). The 2VO-induced low gamma reduction during exploration was significantly rescued by the 40 Hz light flicker compared between 2VO + LED and 2VO (RM-2ANOVA with Sidak's post hoc test, $P_{3d} < 0.0001$, $P_{7d} < 0.0001$, $P_{14d} < 0.0001$, Fig. 1o, p). Therefore, 40 Hz light flicker rescued 2VO-induced impairment in CA1 low gamma and theta-low gamma PAC.

**30–50 Hz light flicker conferred CA1 neuroprotection.** To demonstrate that 40 Hz light flicker confers neuroprotection, the 2VO mouse model was empirically established and characterized to show selective CA1 damage. A protocol of 5 min 2VO reduced hippocampal regional blood flow (hrCBF) to 85%–95% (Supplementary Fig. 1a, b) with simultaneous hypotension at 35–40 mmHg during 2VO (Supplementary Fig. 1c). Strong Fluoro-Jade B (FJB) fluorescence occurred in the CA1 area at 3–14 d after 2VO (Fig. 2). No FJB neurons were in the CA2, CA3, cortex, striatum, and thalamus nuclei (Supplementary Fig. 2a). The numbers of parvalbumin-positive (PV+) interneurons, as shown in Supplementary Fig. 2(b, c), were also not significantly different compared amongst the four experimental groups at 3 d, 7 d, and 14 d post 2VO, indicating selective injury to CA1 excitatory

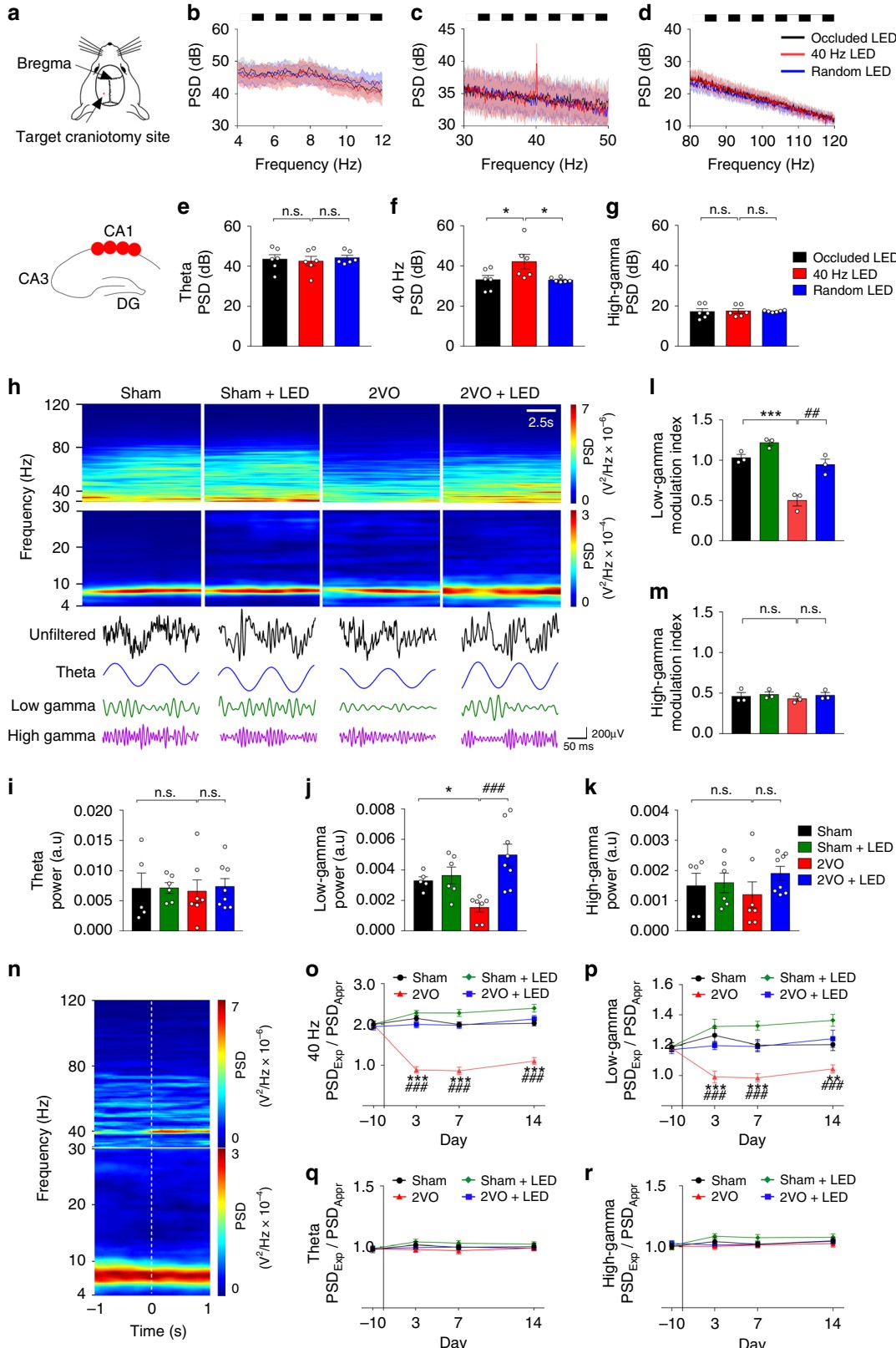

neurons. 40 Hz light flicker treatment significantly reduced CA1 FJB positive neurons in the 2VO + LED group (2ANOVA: $F_{(1, 30)} = 23.75$, $P < 0.0001$, with LSD's post hoc test for 2VO vs. 2VO + LED: $P_{3day} = 0.0014$, $P_{7day} = 0.0072$, $P_{14day} = 0.0495$, Fig. 2c). At 14 d post 2VO, FJB positive cells were not present in the CA1 region of mice treated with 40 Hz LED light (2VO + LED group,

Fig. 2b, c), demonstrating remarkable neuroprotection to CA1 neurons by 40 Hz light flicker.

Delayed light treatment at 3 d after 2VO was not neuroprotective compared with the early treatment group (2 h after 2VO) (Two-tailed unpaired t-test, $t_{(9)} = 2.317$, $P = 0.0457$, Fig. 2b, the last column on the right and Fig. 2d). Reparably, light flickers at

**Fig. 1 Recording of CA1 LFP in behaving mouse. a** A multielectrode array was dorsally implanted into the CA1 7 d before 2VO experimentation. Recordings of LFP from the dorsal hippocampal CA1 region in vivo were performed on free-moving mice. **b–d**, LFP PSD of CA1 during light flicker. **e–g** Quantification of LFP PSD of the three groups of mice with the same legend shown in **g** ($n = 6$ sessions/5 mice each group). **h** Example spectrograms for the four groups of mice recorded 10 s in CA1 during low-speed free walking. The excerpt of raw LFPs of theta, low gamma, and high gamma oscillations are shown in the lower panel of **h**, and the PSD intensity is color-coded according to the color scale shown on the right. **i–k** Quantification of the LFP powers of the four groups of mice in the theta, low gamma, and high gamma frequencies. The same group legend is shown in **k** (Sham: $n = 5$ sessions/3 mice; Sham+LED: $n = 6$ sessions/3 mice; 2VO: $n = 7$ sessions/3 mice; 2VO + LED; $n = 8$ sessions/3 mice). **l,m** PAC analysis of low gamma and high gamma, respectively, to the theta phase as indicated by the modulation index ($n = 3$ mice). **n** Example of moving window spectrogram of hippocampal CA1 LFP time-locked to the initiation of approach (<0 s) and explore (>0 s) a novel object and the PSD intensity is color-coded according to the color scale shown on the right. **o–r** The ratio of PSD between Exploration ($PSD_{Exp}$) and Approach ($PSD_{Appr}$) of the four groups of mice at the indicated oscillation frequencies on the y-axis ($n = 6$ sessions/3 mice). Data represent the mean ± SEM. Error bars indicate SEM. *$P < 0.05$, ##$P < 0.01$, *** and ###$P < 0.001$; The test used in **e–g** 1ANOVA with LSD's post hoc analysis; **i–k** 2ANOVA with LSD's post hoc test; **l**, **m** 2ANOVA with Tukey's post hoc test; **o–r** RM-2ANOVA with Sidak's post hoc test. Source data are provided as a Source Data file.

30 Hz, 40 Hz, and 50 Hz all showed significant protection to CA1 pyramidal neurons compared with the arrhythmic frequency (1ANOVA, $F_{(3, 16)} = 5.63$, $P = 0.0079$, with Dunnett's post hoc test at $P_{30Hz \text{ vs. Random}} = 0.0106$, $P_{40Hz \text{ vs. Random}} = 0.0077$, $P_{50Hz \text{ vs. Random}} = 0.0076$, Fig. 2e). In contrast, the 10 Hz, 80 Hz, and random frequency light flickers were not protective based on FJB staining of hippocampal brain sections (1ANOVA, $F_{(2, 11)} = 0.02985$, $P = 0.9707$, Dunnett's post hoc test: $P_{10Hz \text{ vs. Random}} = 0.9971$, $P_{80Hz \text{ vs. Random}} = 0.9617$, Fig. 2e). These data suggested that light flicker protection was time- and frequency-dependent.

Several factors were investigated to understand how light flickers protect neurons. First, the cerebral blood flow (CBF) was not affected by light flicker. The laser Doppler flowmetry of hrCFB showed no statistical differences amongst the four experimental groups (Supplementary Fig. 1b). The CBF of the whole cortical area in general (Supplementary Fig. 1e, f), and the visual cortex area in particular (Supplementary Fig. 1e, g), were also not significantly different at 3 d, 7 d, and 14 d between 2VO and 2VO + LED groups, and between Sham and Sham+LED groups, respectively, as measured using laser speckle contrast imaging. FITC-lectin perfusion staining at the end of 14 d of light treatment also showed that blood vessel densities in the hippocampal and visual cortex areas were not significantly different between 2VO and 2VO + LED groups (VC: 2ANVOA $F_{(1, 40)} = 0.0559$, $P = 0.8143$; HPC: $F_{(1, 38)} = 0.1657$, $P = 0.6863$, Supplementary Fig. 1h-i). Second, 2VO activated microglia. However, the Sham and 2VO groups showed no increase in microglia in response to light flicker (Supplementary Fig. 2d–h), indicating an inverse correlation with neuroprotection. The numbers of GFAP positive astrocytes were similar compared amongst all four groups of mice (Supplementary Fig. 3). These data indicated that light flicker might have a direct effect on neurons.

**40 Hz light flicker improved cognitive functions of 2VO mice.** A battery of cognitive-behavioral tests was performed (Fig. 3a). The open-field test (OFT) showed that 2VO mice performed poorly compared with the Sham group in terms of the total distance traveled (2ANOVA, $F_{(1, 28)} = 8.338$, $P = 0.0074$, Tukey's post hoc test, $P_{Sham \text{ v.s. } 2VO} = 0.0308$, Fig. 3b), the distance in the field center ($F_{(1, 28)} = 5.837$, $P = 0.0225$; Tukey's post hoc test, $P_{Sham \text{ vs. } 2VO} = 0.0477$, Fig. 3c) and the percentage of time spent in the center ($F_{(1, 28)} = 12.05$, $P = 0.0017$; Tukey's post hoc test, $P_{Sham \text{ vs. } 2VO} = 0.0480$, Fig. 3d). Light flicker treatment enhanced OFT performance of 2VO group (Tukey's post hoc test for 2VO vs. 2VO + LED, $P = 0.0458$ in Fig. 3b; $P = 0.0223$ in Fig. 3c; $P = 0.0015$ in Fig. 3d), demonstrating that 40 Hz light flicker enhanced the locomotor ability, but did not produce anxiety-like behaviors.

The water maze test showed that 40 Hz light flicker significantly attenuated 2VO-induced impairments in spatial leaning ability and reference memory (Fig. 3e–h). All four groups of mice showed no difference in swimming speed (Fig. 3g). However, the escape latency and pathlength of 2VO mice were significantly longer compared with the Sham group, respectively, during the acquisition test (Escape latency: RM-2ANOVA, $F_{(1, 15)} = 14.52$, $P = 0.0017$, LSD's post hoc test, $P_8 = 0.579$, $P_9 = 0.017$, $P_{10} = 0.016$, $P_{11} = 0.011$, $P_{12} = 0.001$, Fig. 3e; Pathlength: $F_{(1, 15)} = 7.118$, $P = 0.0175$, LSD's post hoc test, $P_8 = 0.652$, $P_9 = 0.162$, $P_{10} = 0.0505$, $P_{11} = 0.007$, $P_{12} = 0.015$, Fig. 3f). A lesser time spent in the target quadrant in the probe test was an indication of poor spatial learning and memory abilities, which was associated with the deficits of object-location associative memory encoding caused by 2VO (2ANOVA with LSD's post hoc test, $P_{Sham \text{ vs. } 2VO} = 0.0438$, Fig. 3h). 40 Hz light flicker significantly enhanced the acquisition abilities of spatial learning and memory compared with the 2VO group (RM-2ANOVA, $F_{(1, 17)} = 10.24$, $P = 0.0053$, LSD's post hoc test, $P_8 = 0.838$, $P_9 = 0.179$, $P_{10} = 0.041$, $P_{11} = 0.034$, $P_{12} = 0.00017$, Fig. 3e; 2ANOVA, $F_{(1, 26)} = 5.492$, $P = 0.0270$, LSD's post hoc test, $P_{2VO \text{ vs. } 2VO+LED} = 0.0107$, Fig. 3h). These data are strong indications that 40 Hz light flicker mitigated cognitive deficits caused by 2VO.

**Enhancement of CA3-CA1 LTPs by 40 Hz light flicker.** There is a strong correlation of theta-gamma PAC strength with LTP and learning memory[27,28]. Experiments were, therefore, designed to determine whether 40 Hz light flicker enhances the fast excitatory synaptic LTPs of the CA3-CA1 network, including NMDA receptor-dependent LTP ($LTP_{NMDAR}$) and L-type voltage-gated calcium channel-dependent LTP ($LTP_{L-VGCC}$), both are important in cerebral ischemia-induced cognitive impairments[18,29]. In the presence of NMDAR antagonist APV, $LTP_{L-VGCC}$ can be induced with 200 Hz tetanic stimulation[30], which served as a pharmacological approach to isolate the two LTPs. The input/output (I/O) curves of field excitatory postsynaptic potentials (fEPSP) amplitude and slope with a series of increasing stimulation intensities were recorded and shown in Fig. 4a, b, respectively. The basal synaptic transmission of 2VO mice was significantly impaired compared with the Sham group in slope (Fig. 4a, RM-2ANOVA, $F_{(1, 4)} = 29.97$, $P = 0.0054$ with Sidak's post hoc test at $P_{0.1} = 0.8291$, $P_{0.2} = 0.0250$, $P_{0.3} < 0.0001$, $P_{0.4} < 0.0001$, $P_{0.5} < 0.0001$, $P_{0.6} < 0.0001$), and in amplitude (Fig. 4b, RM-2ANOVA, $F_{(1, 4)} = 37.28$, $P = 0.0036$ with Sidak's post hoc test at $P_{0.1} = 0.5231$, $P_{0.2} < 0.0001$, $P_{0.3} < 0.0001$, $P_{0.4} < 0.0001$, $P_{0.5} < 0.0001$, $P_{0.6} < 0.0001$). The other three groups of mice were not significantly different.

The 2VO group showed a significant lack of $LTP_{NMDAR}$ compared to the Sham group at 3 d (RM-2ANOVA, $F_{(1, 4)} = 578.70$, $P < 0.0001$, Fig. 4d; 2ANOVA with Tukey's post hoc test,

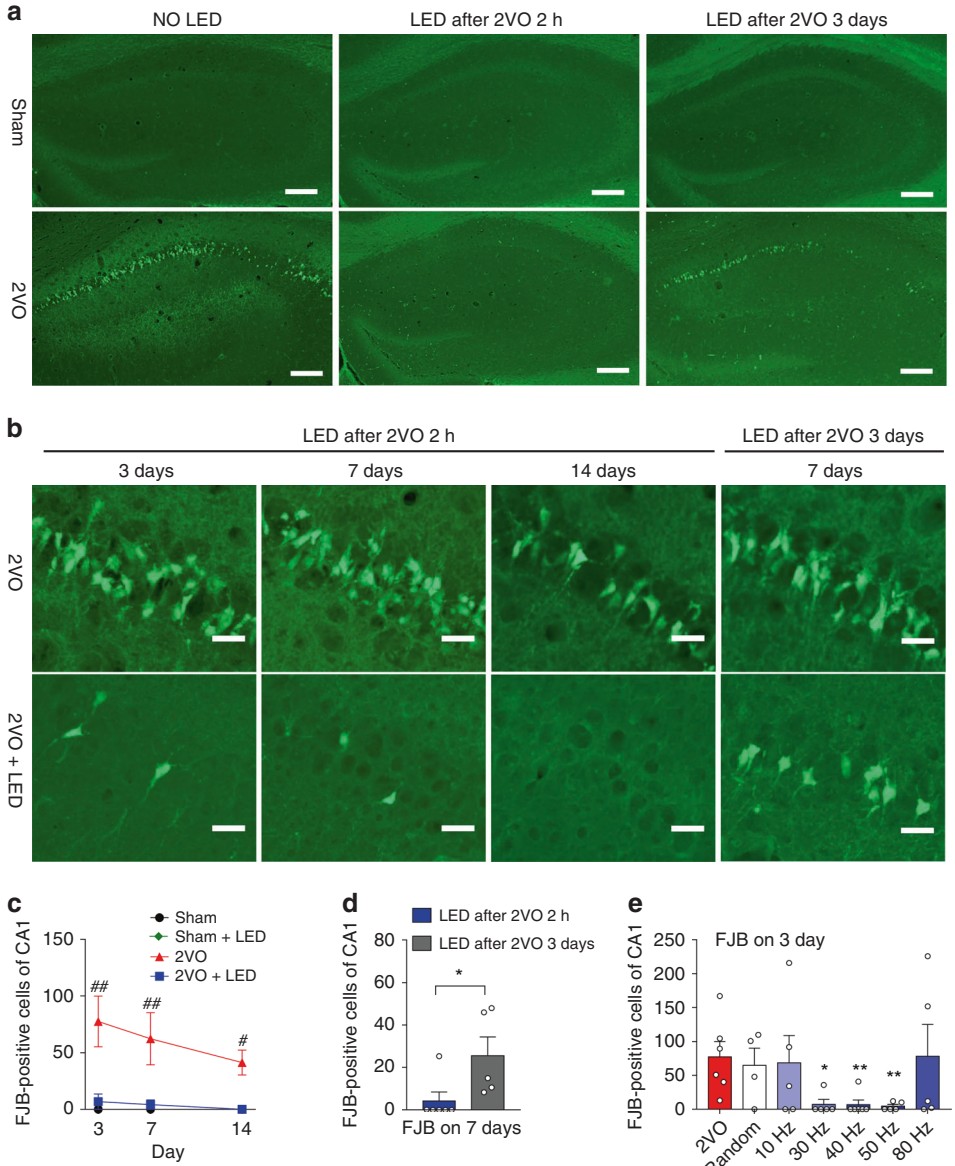

**Fig. 2 Protection to CA1 neurons conferred by 40 Hz light treatment. a** Coronal brain sections were stained with FJB. Hippocampal areas were shown in **a** highlighting 2VO-induced CA1 degenerating neurons (bright green colored cells). Scale bars = 50 μm. **b** Enlarged CA1 area from 3 d, 7 d, and 14 d after 2VO with or without 40 Hz light treatment. Scale bars = 20 μm. The last column of **b** showed 40 Hz light treatment at 3 d after 2VO. **c** and **d** Quantification of CA1 FJB positive neurons within CA1 ($n$ = 6 mice each group in **c**; LED after 2VO 2 h: $n$ = 6 mice; LED after 2VO 3d: $n$ = 5 mice). **e** 2VO mice were treated with different rhythmic frequencies, and arrhythmic frequency of lights and FJB staining was performed 3 d after 2VO. $n$ = 6 mice in 2VO and 40 Hz groups; $n$ = 4 mice in Random group; $n$ = 5 mice in other groups. Data represent the mean ± SEM. Error bars indicate SEM. ## indicates a comparison between 2VO and 2VO + LED group. *indicates a comparison between 2VO + LED group received 40 Hz light 2 h or 3 d after 2VO surgery, and ** in panel **e** indicates a comparison between the random group and 2VO mice received different frequencies of light treatment (*,#$P$ < 0.05, and **$P$ < 0.01). The test used in **c**: 2ANOVA with LSD's post hoc analysis; **d** Two-tailed unpaired t-test; **e** 1ANOVA with Dunnett's post hoc test. Source data are provided as a Source Data file.

$P$ < 0.0001, Fig. 4e), 7 d and 14 d (Supplementary Fig. 4a–c) after 2VO. Light flicker treatment of 2VO mice with 40 Hz, but not 10 Hz, 80 Hz, and random frequency, significantly increased $LTP_{NMDAR}$ in the 2VO + LED group after 3 d (RM-2ANOVA, $F_{(1, 4)}$ = 25.26, $P$ = 0.0074, Fig. 4d; 2ANOVA with Tukey's post hoc test, $P$ = 0.0006, Fig. 4e and Supplementary Fig. 4e, f), and 7d and 14 d (Supplementary Fig. 4a–c), demonstrating enhanced CA3-CA1 excitatory synaptic strength after 40 Hz light stimulation.

$LTP_{L-VGCC}$ was pharmacological isolated using D-APV (50 μM) or D-APV + nifedipine (10 μM) (Supplementary Fig. 4d, g).

A near-complete loss of $LTP_{L-VGCC}$ occurred in 3 d 2VO mice compared with the Sham group (RM-2ANOVA, $F_{(1, 4)}$ = 8.917, $P$ = 0.0405, Fig. 4f; 2ANOVA with Tukey's post hoc test, $P$ = 0.0059, Fig. 4g). In contrast, 40 Hz light flicker restored $LTP_{L-VGCC}$ to the normal level at 3 d when compared with 2VO group (RM-2ANOVA, $F_{(1, 5)}$ = 162.6, $P$ < 0.0001, Fig. 4f; 2ANOVA with Tukey's post hoc test, $P$ = 0.0020, Fig. 4g). 2VO mice treated with 10 Hz, 80 Hz, or random frequency light flicker failed to restore 2VO-induced reduction in $LTP_{L-VGCC}$ (Supplementary Fig. 4h, i). These data demonstrated that 40 Hz light flickers enhanced postsynaptic LTPs.

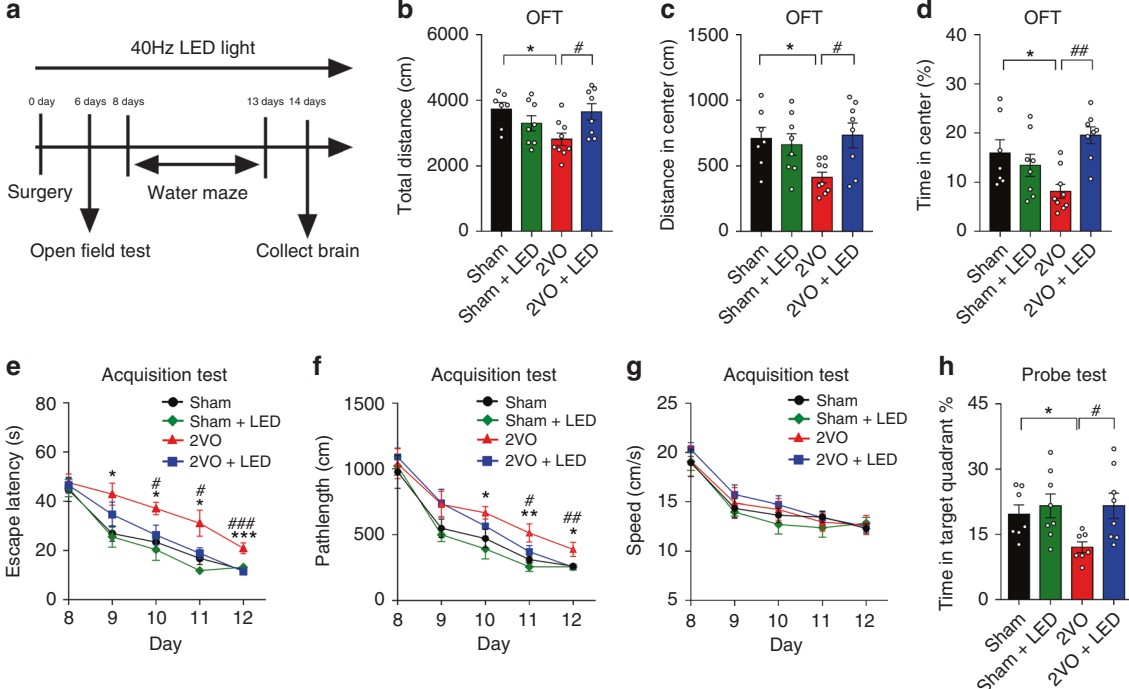

**Fig. 3 Improvement of cognitive functions of 2VO mice after 40 Hz light visual stimulation. a** A schema showing experimental design and timeline. **b–d** OFT results of four groups of mice. Total distance traveled (**b**), distance in the center (**c**), and percentage of time spent in the center (**d**) were measured and compared amongst the four experimental groups (Sham: $n = 7$; Sham+LED: $n = 8$; 2VO: $n = 9$; 2VO + LED: $n = 8$ in OFT). **e** Water maze test (WMM) was also performed on the four experimental groups with days indicated. Escape latency to the platform (**e**) was measured during the acquisition test at 8, 9, 10, 11, and 12 d after 2VO surgery. Pathlength (**f**) and locomotion speed (**g**) of the four groups were also recorded and plotted (Sham: $n = 8$; Sham+LED: $n = 8$; 2VO: $n = 9$; 2VO + LED: $n = 10$ in the acquisition test of MWM). **h** The percentage of time spent in the target quadrant during the spatial exploring stage of the MWM probe test (Sham: $n = 7$; Sham+LED: $n = 8$; 2VO: $n = 7$; 2VO + LED: $n = 8$ in the probe test of MWM). Data were mean ± SEM. Error bars indicated SEM. * indicates a statistical difference between Sham and 2VO group; # indicates a statistical difference between 2VO and 2VO + LED group. *, #$P < 0.05$, **, ##$P < 0.01$, ***, ###$P < 0.001$; Test used in **b–d** 2ANOVA with Tukey's post hoc test; **e–g** RM-2ANOVA with LSD's post hoc test; **h** 2ANOVA with LSD's post hoc test. Source data are provided as a Source Data file.

**40 Hz light flicker enhanced CA3-CA1 presynaptic plasticity.** To further understand the effect of light flicker on CA3-CA1 synaptic transmission, presynaptic short-term plasticity was measured using the paired-pulse ratio (PPR), which is inversely related to the initial presynaptic vesicle release probability[31,32]. Control experiments were performed to show that neither D-APV nor nifedipine affected CA3-CA1 PPR, since both target postsynaptic receptors (Supplementary Fig. 5a, b). The 2VO group significantly elevated PPR compared to the Sham group (2ANOVA, $F_{(1, 15)} = 26.5$, $P = 0.0001$, Tukey' post hoc test $P_{Sham\ vs.\ 2VO} = 0.0009$, Fig. 5b), indicating the reduced readily release probability of neurotransmitters. 40 Hz light treatment restored presynaptic readily release probability and enhanced presynaptic short-term plasticity as evidenced by the reduction of 2VO group PPR (2ANOVA, $F_{(1, 15)} = 37.01$, $P < 0.0001$, Tukey' post hoc test $P_{2VO+LED\ vs.\ 2VO} = 0.0003$, Fig. 5b). In contrast, stimulation with arrhythmic light, 10 Hz, and 80 Hz light did not affect 2VO group PPR severing as negative controls (Supplementary Fig. 4j, k).

FM1-43, a fluorescent marker of presynaptic activity, was used to directly visualize the dynamic changes of presynaptic vesicle release probabilities due to 40 Hz light treatment. Imaging of presynaptic vesicle exocytosis at the presynaptic boutons of CA3-CA1 excitatory synapses in acute hippocampal slices was performed using two-photon excitation microscopy (Fig. 5c–e). The brief loading protocol used here selectively labeled presynaptic terminals (Fig. 5c)[33]. The puncta shown in Fig. 5c represent glutamatergic presynaptic terminals from CA3 pyramidal neurons, which accounted for most of the terminals examined in this region. During a train of 1.5 Hz stimuli, the

intensity of puncta staining decayed with an approximately exponential time course (Fig. 5d), reflecting the first-order kinetics of vesicle release.

The rate of FM1-43 unloading, shown as the inverse halftime of intensity decay ($1/t_{1/2}$), in 2VO CA3 glutamatergic presynaptic terminals was significantly decreased compared with the Sham group (2ANOVA, $F_{(1, 9)} = 23.26$, $P = 0.0009$, Tukey's post hoc test $P_{Sham\ vs.\ 2VO} = 0.0006$, Fig. 5e). The 2VO + LED group completely restored the unloading kinetics to the levels of the Sham group (2ANOVA with Tukey's post hoc test, $P_{2VO\ vs.\ 2VO+LED} = 0.0006$, Fig. 5e), providing additional evidence to support the PPR findings. In contrast, stimulation with arrhythmic light, 10 Hz and 80 Hz light flicker did not affect 2VO group unloading kinetics, severing as negative controls (Supplementary Fig. 4l, m).

**40 Hz light flicker increased 2VO mice dendritic plasticity.** The dendritic spine formation and alterations were examined under two-photon excitation microscopy using transgenic Thy1-YFP male mice. Dendritic spines of YFP-expressing CA1, CA3 pyramidal neurons were examined in hippocampal brain slices using the Sholl analysis[34] (Fig. 6a, b). A Z-stack of over 500 µm optical scans of the CA3-CA1 area were compiled, and individual pyramidal neurons of interests were highlighted using camera lucida tracing (Fig. 6c). Three types of dendritic spines were identified based on their shapes, including stubby, mushroom, and thin (Fig. 6d). More than 95% of them were stubby or mushroom-shaped boutons (Fig. 6d lower panel). At 3 d after 2VO, a significant reduction in total number of spines (2ANOVA, $F_{(1, 20)} = 23.45$, $P < 0.0001$, Tukey's

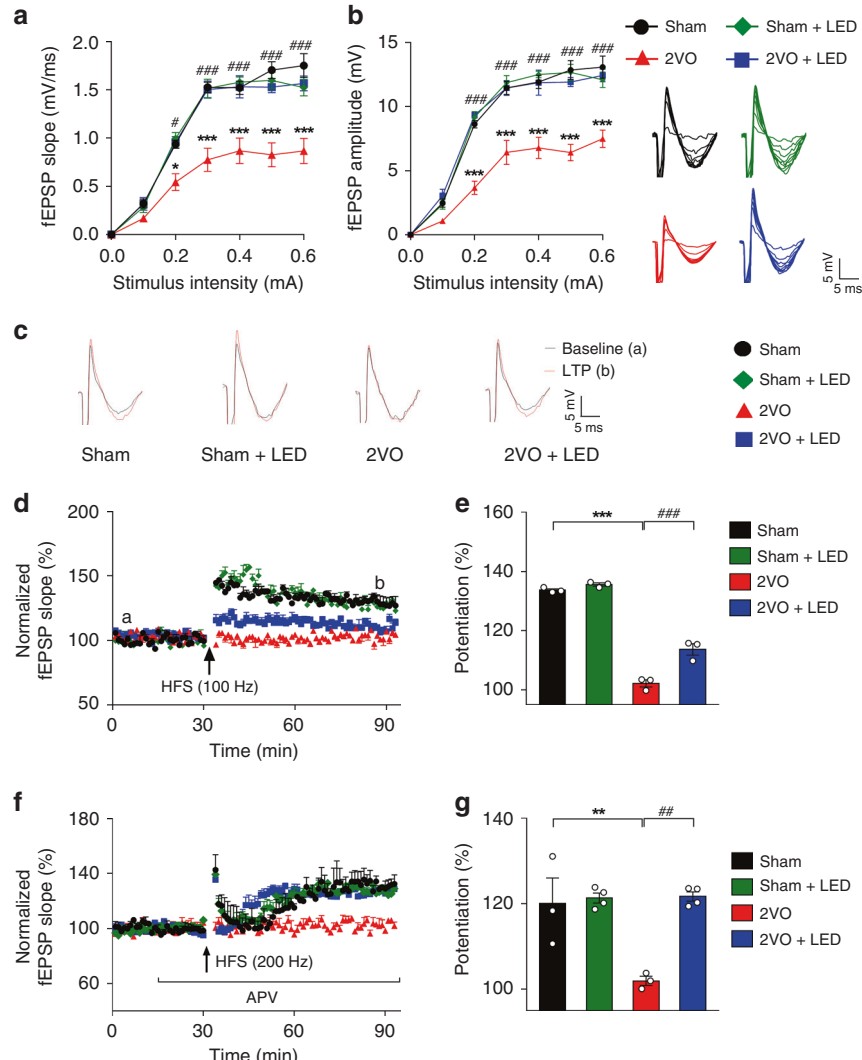

**Fig. 4 Increased LTPs in response to 40 Hz light treatment of 2VO mice.** The input/output curve (I/O) of fEPSP slope (**a**) and amplitudes (**b**) across stimulation intensities of all four groups of mice ($n = 3$). **c** The $LTP_{NMDAR}$ was induced by 100 Hz frequency tetanic stimulation of the Schaffer collaterals projected to the CA1 pyramidal neurons on acute hippocampal slices 3 d after 2VO (**d**). **e** The first 30 min of evoked responses were normalized and used as the baseline responses of $LTP_{NMDAR}$. The magnitude of $LTP_{NMDAR}$ was determined according to the responses between 0 and 60 min after the HFS ($n = 3$). **f** $LTP_{L-VGCC}$ was induced by stimulation of the Schaffer collaterals with 200 Hz tetanic stimulation + D-APV (50 µm) throughout the experiment. **g** The first 30 min of evoked responses were normalized and used as the baseline responses of $LTP_{L-VGCC}$. $LTP_{L-VGCC}$ magnitude was determined according to the responses between 0 and 60 min after the HFS (Sham $n = 3$ mice, Sham+LED $n = 4$ mice, 2VO $n = 3$ mice, 2VO + LED $n = 4$ mice). Data represent the mean ± SEM. Error bars indicate SEM. * indicates a comparison between Sham and 2VO group; # indicates a comparison between 2VO and 2VO + LED group. *, #$P < 0.05$, **, ##$P < 0.01$, ***, ###$P < 0.001$. The test used in **a**, **b**, **d**, **f** RM-2ANOVA with Sidak's post hoc test. **e**, **g** 2ANOVA with Tukey's post hoc test. Source data are provided as a Source Data file.

post hot test, $P_{Sham\ vs.\ 2VO} < 0.0001$, Fig. 6e) and stubby shaped spines (2ANOVA, $F_{(1, 20)} = 45.59$, $P < 0.0001$, Tukey's post hot test, $P_{Sham\ vs.\ 2VO} < 0.0001$, Fig. 6i) occurred. These spines were on the primary and secondary branches of the apical dendrites of CA1 pyramidal neurons. 2VO + LED mice showed significantly increased spine numbers on CA1 apical dendrites (2ANOVA with Tukey's post hoc test, $P_{2VO\ vs.\ 2VO+LED} < 0.0001$, Fig. 6e), particularly the stubby-shaped spines (2ANOVA with Tukey's post hoc test, $P_{2VO\ vs.\ 2VO+LED} < 0.0001$, Fig. 6i), indicating enhanced CA3-CA1 synaptic plasticity. It is worth noting that at 3 d post 2VO, CA3 neurons were intact with no significant loss of dendritic spines (2ANOVA, Fig. 6f, $F_{(1, 20)} = 0.738$, $P = 0.4005$; Fig. 6h, $F_{(1, 20)} = 0.1321$, $P = 0.7200$; Fig. 6j, $F_{(1, 20)} = 1.479$, $P = 0.2381$; Fig. 6l, $F_{(1, 20)} = 0.2157$, $P = 0.6473$).

To demonstrate the long-term effects of 40 Hz light flicker on CA1 dendritic spines, CA1 and CA3 apical and basal dendritic spines were examined at 3 d, 7 d, and 14 d post 2VO. 2VO caused a significant loss of CA1 dendritic spines throughout the 14 d periods (Supplementary Fig. 6a, b), while 40 Hz light treatment prevented the loss of CA1 dendritic spines (Supplementary Fig. 6a, b). In contrast, the loss of CA3 dendritic spines only appeared at 7 d and 14 d post 2VO (Supplementary Fig. 6c, d), and such a loss was completely rescued in the 2VO + LED group (Supplementary Fig. 6c, d), indicating that loss of connection with CA1 neurons may eventually lead to CA3 dendritic spine loss in the absence of 40 Hz light treatment. Together, these results demonstrated that 40 Hz flicker light strengthened CA3-CA1 synaptic connections.

**40 Hz light flicker altered hippocampal protein expression.** To identify specific molecular pathways involved in CA3-CA1

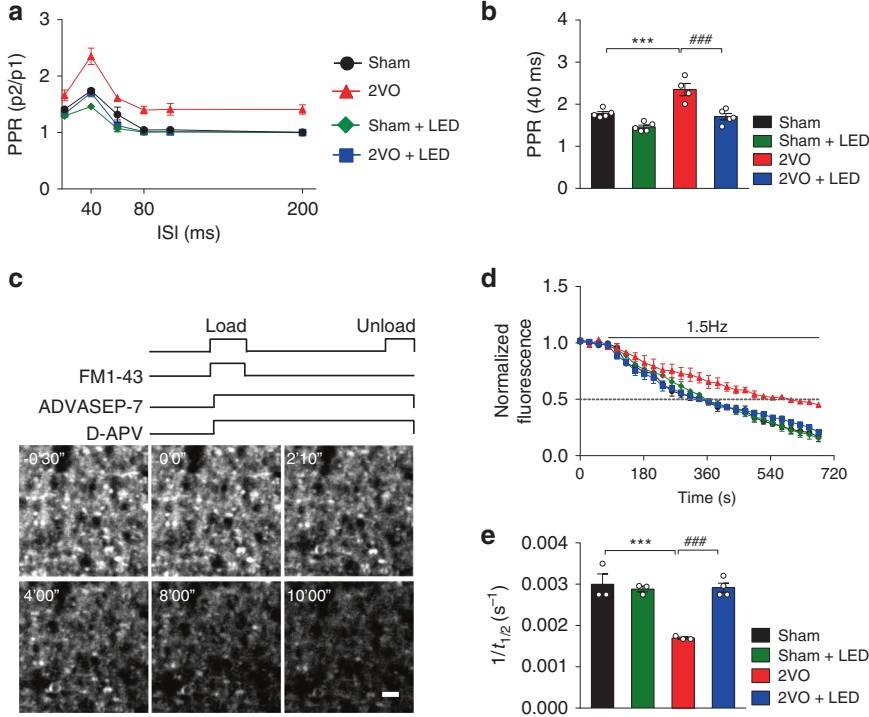

**Fig. 5 40 Hz light stimulation enhanced CA3-CA1 presynaptic short-term plasticity. a** Hippocampal slices from the Sham, Sham+LED, 2VO, and 2VO + LED groups at 3 d after the 2VO surgery were subjected to PPR measurements. The inter-stimulus intervals (ISI) were from 20–200 ms. The ratios of the slope of the second response to that of the first at the 40 ms ISI for **a** were measured and plotted in **b** (Sham: $n = 5$ mice; Sham+LED: $n = 5$ mice; 2VO: $n = 4$ mice; 2VO + LED: $n = 5$ mice). **c** The protocol for determining the kinetics of FM1-43 release from presynaptic terminals. Presynaptic terminals were labeled by exposure to 5 μM FM1-43 (numbers indicate time in minutes and seconds; zero time represents the beginning of the unloading stimulation; Scale bar = 2 μm). The first image was taken before unloading stimulation. **d** Normalized fluorescence intensity decay of puncta to the initial FM1-43 loading intensity were plotted. Each point represents the average of a total of 10–15 boutons from different slices of mice. The rate of puncta unloading in brain slices ($1/t_{1/2}$) was plotted and shown in **e** (Sham: $n = 3$ mice; Sham+LED: $n = 3$ mice; 2VO: $n = 3$ mice; 2VO + LED: $n = 4$ mice). Data represent the mean ± SEM. Error bars indicate SEM. ***, ###$P < 0.001$. The test used in **b**, **e** 2ANOVA with Tukey' post hoc test. Source data are provided as a Source Data file.

network responses to 2VO and 40 Hz light treatment, the whole hippocampi from 2VO, 2VO + LED, and the Sham group 3 d post-surgery were isolated and subjected to proteomic analysis (Fig. 7). Using bioinformatic gene ontology analysis, 2VO altered the expression of more than 60 proteins over 1.2-fold changes compared with the Sham group, and 31 of those with altered expression in the 2VO + LED group (Supplementary Fig 7). The identified protein groups were related to synaptic functions (45%), cell death (24%), and transcription and translation (14%) (Fig. 7a). The remainders were related to cell differentiation (5%), epigenetic regulations (5%), immune response (3%), oxidative stress (2%), and mitochondrial functions (2%). We focused on genes potentially involved in modulating synaptic neurotransmission, such as the regulator of G-protein signaling 12 (RGS12), a regulator of the N-VGCC[35]. The expression level of RGS12, shown by western blotting (Fig. 7b), was down-regulated by 20% and 30% in 2VO group at 3 d and 7 d, respectively, compared with Sham group (2ANOVA, $F_{(1, 8)} = 5.3$, $P = 0.0503$, LSD' post hot test, $P_{\text{Sham vs. 2VO}} = 0.0263$, Fig. 7c, d), and up-regulated by 60% and 35% at 3 d and 7 d 2VO + LED group, respectively, compared with the 2VO group (2ANOVA, $F_{(1, 8)} = 5.53$, $P = 0.0466$, LSD' post hot test, $P = 0.0249$, Fig. 7c, d).

**Increased RGS12 modulation of N-VGCC short-term plasticity.** To demonstrate RGS12 involvement in presynaptic plasticity in response to 40 Hz light flicker, two experimental approaches were used: a decoy peptide blocking the interaction of endogenous RGS12 with N-VGCC, and an AAV-shRNA knocking down

RGS12 expression in the 2VO brain (Fig. 8). The decoy RGS12 peptide GK23 has an identical peptide sequence to that of the N-VGCC centered around Tyr-804. The GK23 sequence was completely lacking in L-VGCC, confirming the specificity of GK23 (Supplementary Fig. 8a). GK23 significantly increased PPR compared with the control peptide GK13 (Supplementary Fig. 5c, d), demonstrating its effectiveness. RGS12 decoy peptide GK23 was delivered to the left cerebroventricle using an Alzet® osmotic pump followed by 2VO surgery and 40 Hz light treatment. As shown in Fig. 8(a, c), GK23, and ω-conotoxin GVIA significantly enhanced PPR of the 3 d 2VO + LED group compared with the GK13-treated 2VO + LED group (1ANOVA, $F_{(2, 6)} = 5.559$, $P = 0.0431$, Dunnett's post hoc test, $P_{\text{GK23 vs. GK13}} = 0.0351$, $P_{\text{GK23 vs. ω-conotoxin GVIA}} = 0.7761$, Fig. 8c left), confirming RGS12 regulation of 40 Hz light-induced presynaptic plasticity. FM1-43 was also used to directly image presynaptic vesicle exocytosis. Acute hippocampal slices from the above-treated 2VO + LED mice were used. The halftime of puncta intensity decay ($1/t_{1/2}$) was significantly reduced in GK23, and ω-conotoxin GVIA treated 2VO + LED mice (1ANOVA, $F_{(2, 6)} = 13.35$, $P = 0.0062$, Dunnett's post hoc test, $P_{\text{GK23 vs. GK13}} = 0.0052$, $P_{\text{GK23 vs. ω-conotoxin GVIA}} = 0.6388$, Fig. 8f left). Furthermore, GK23 and ω-conotoxin GVIA completely inhibited the occurrence of postsynaptic LTP$_{\text{NMDAR}}$ and LTP$_{\text{L-VGCC}}$ in the 2VO + LED group (Fig. 8g: RM-2ANOVA, GK13 vs. GK23: $F_{(1, 4)} = 34.5$, $P = 0.0042$, GK23 vs. ω-conotoxin GVIA: $F_{(1, 4)} = 0.001461$, $P = 0.9713$. Figure 8i left: 1ANOVA with Dunnett's post hoc test $P_{\text{GK23 vs. GK13}} = 0.0013$, $P_{\text{GK23 vs. ω-conotoxin GVIA}} = 0.9985$. Figure 8j: RM-

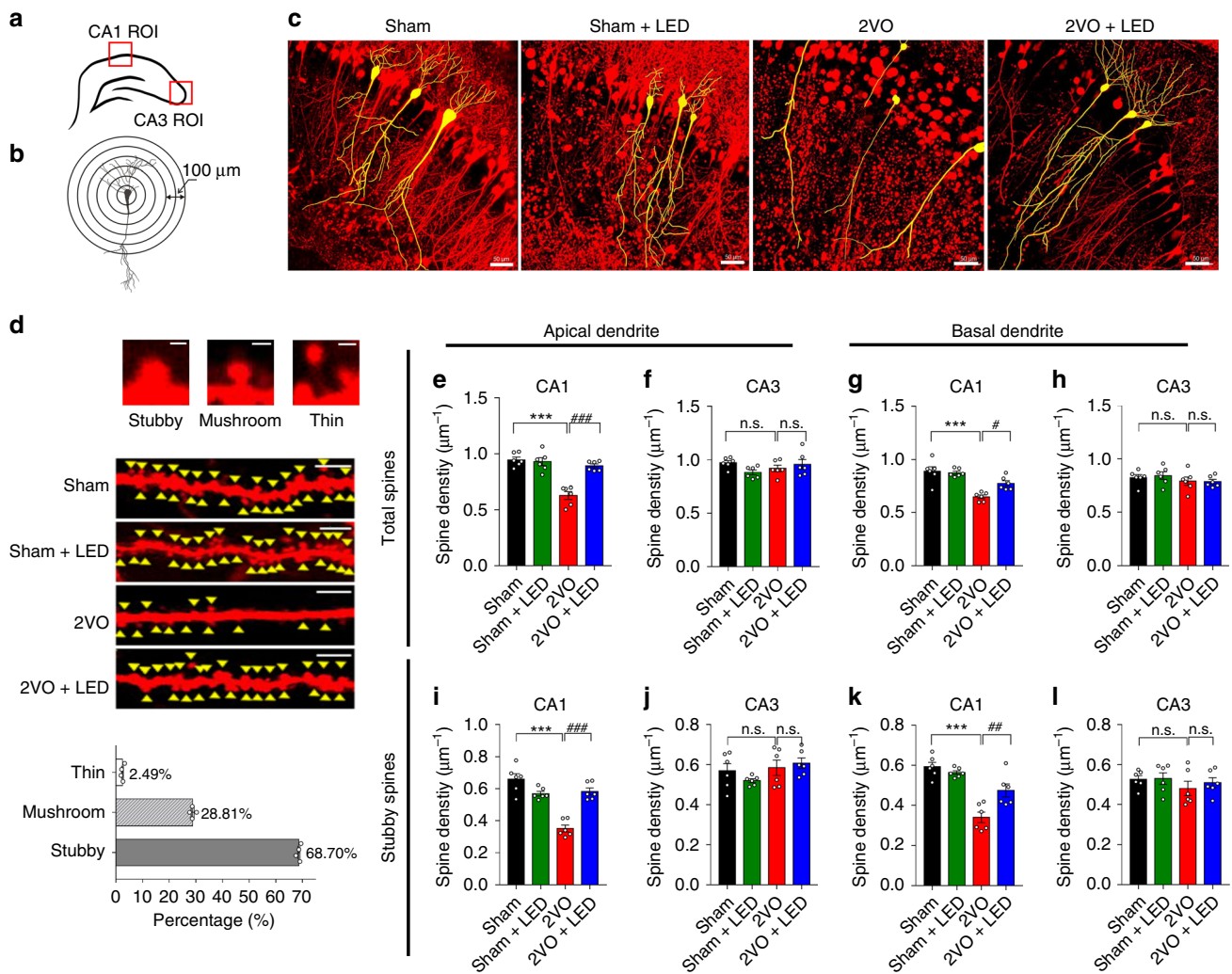

**Fig. 6 Structural plasticity of CA1 and CA3 neurons in response to 40 Hz light treatment. a** A diagram of CA1 and CA3 areas where two-photon images were taken and shown in panel **c**. **b** Camera lucida of a CA1 pyramidal neuron for Sholl analysis. Scale bar = 100 μm. **c** Representative pyramidal neurons in CA1 of four groups (yellow colored lines) were put together using a z-stack of over 500 optical sections (6 repeats from $n = 3$ mice in each group). Scale bars = 50 μm. **d** Representative two-photon images of stubby, mushroom, and thin-formed dendritic spines (top panels) of the apical dendrites of Thy1-YFP mice of Sham, Sham+LED, 2VO, and 2VO + LED groups at 3 d post-surgery (6 repeats from $n = 4$ mice). Scale bars = 5 μm. The percentage of stubby, mushroom and thin shaped boutons were averaged and shown in the lower panel of **d** ($n = 4$ mice). **e–l** Quantifications of the number of dendritic spines per unit length of the CA1 and CA3 apical and basal dendrites, respectively, at 3 d post-surgery ($n = 6$ slices/3 mice in each group). Data represent the mean ± SEM. Error bars indicate SEM. *indicates differences between the Sham group and 2VO group; # indicates differences between the 2VO group and 2VO + LED group. #$P < 0.05$, ##$P < 0.01$, ***, ###$P < 0.001$. The test used in **e–l** 2ANOVA with Tukey's post hoc test. Source data are provided as a Source Data file.

2ANOVA, GK13 vs. GK23: $F_{(1, 5)} = 9.099$, $P = 0.0295$, GK23 vs. ω-conotoxin GVIA: $F_{(1, 5)} = 0.006603$, $P = 0.9384$. Figure 8l left: 1ANOVA with Dunnett's post hoc test $P_{GK23\ vs.\ GK13} = 0.0170$, $P_{GK23\ vs.\ ω-conotoxin\ GVIA} = 0.9945$). These data demonstrated that inhibition of presynaptic RGS12 interaction with N-VGCC suppressed light flicker restoration of presynaptic release probability in 2VO mice.

The second approach was to use AAV-shRNA to knockdown CA1 RGS12 expression. AAV-shRNA-RGS12 and control AAV-shRNA-CTL structures were shown in Supplementary Fig. 8b. The hippocampal CA1 neuron expression of AAV-shRNA-RGS12 for 3 weeks (Supplementary Fig. 8c) successfully reduced the expression level of RGS12 in the hippocampus (Supplementary Fig. 8d). Knocking down RGS12 expression level produced a similar effect to that of the GK23 decoy peptide in terms of the increased CA3-CA1 PPR in 2VO + LED mice (Two-tailed unpaired t-test, $t_{(4)} = 8.197$, $P = 0.0012$, Fig. 8b, c, right), and

reduced presynaptic vesicle release probability (Two-tailed unpaired t-test, $t_{(4)} = 3.011$, $P = 0.0395$, Fig. 8e, f right). AAV-shRNA-RGS12 also significantly reduced the levels of postsynaptic LTP$_{NMDAR}$ and LTP$_{L-VGCC}$ in the 2VO + LED group of mice compared with AAV-shRNA-CTL (Fig. 8h: RM-2ANOVA, $F_{(1, 4)} = 10.4$, $P = 0.0321$. Fig. 8i right: Two-tailed unpaired t-test, $t_{(4)} = 3.225$, $P = 0.0321$. Figure 8k: RM-2ANOVA, $F_{(1, 4)} = 63.5$, $P = 0.0013$. Figure 8l right: Two-tailed unpaired t-test, $t_{(4)} = 7.969$, $P = 0.0013$). Importantly, knocking down RGS12 diminished the rescuing effect of 40 Hz light flicker on CA1 low gamma (Supplementary Fig. 8e, f). In contrast, theta and high gamma oscillations PSD ratios were not affected throughout the 14 d treatment (Fig. S8g, h), demonstrating the important role of RSG12 in mediating 40 Hz light response in the CA1 neurons.

Collectively, these data demonstrated that 40 Hz light flicker conferred neuroprotection through increased RGS12-regulation of the N-VGCC-dependent CA3-CA1 synaptic plasticity.

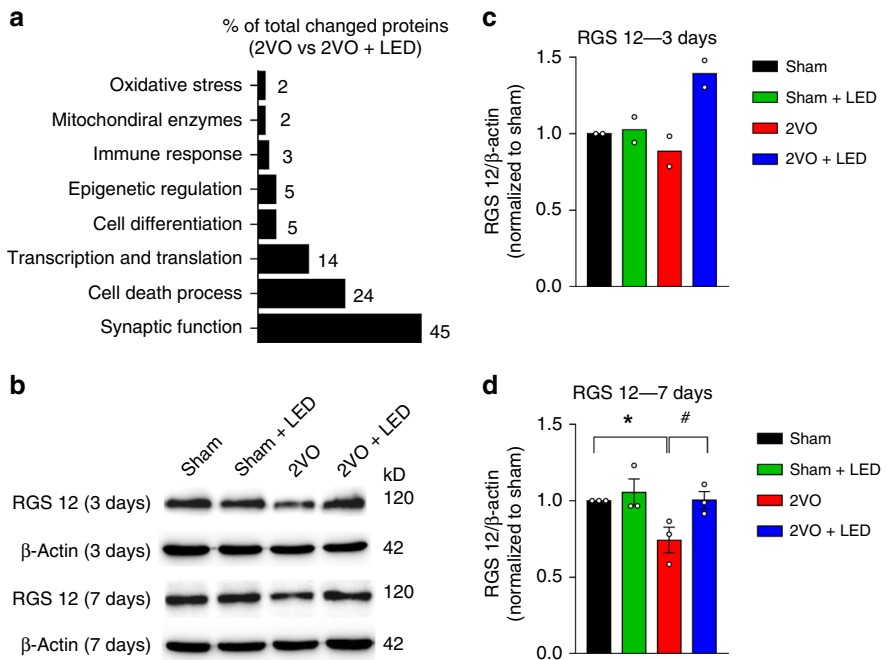

**Fig. 7 Profiling hippocampal protein expression associated with 40 Hz light treatment. a** GO analysis according to the functions of 31 proteins, which significantly altered their expression in the 2VO + LED group. **b** Representative western blots are showing increased levels of RGS12 expression 3 d (4 repeats from $n = 2$ mice) and 7 d (6 repeats from $n = 3$ mice) post 2VO. **c, d** Quantifications of RGS12 expression using β-actin as an internal standard of 3 d ($n = 2$ mice in each group) and 7 d ($n = 3$ mice in each group) post 2VO groups, respectively. Data represent the mean ± SEM. Error bars indicate SEM. * is for comparison between Sham and 2VO groups; # is for comparison between 2VO and 2VO + LED groups. *,#$P < 0.05$. The test used in **d** 2ANOVA with LSD's post hoc test. Source data are provided as a Source Data file.

Enhanced CA3-CA1 synaptic strength provided a survival signal to CA1 neurons leading to neuroprotection during cerebral ischemia.

## Discussion

Data presented here support a causal relationship between hippocampal low gamma and cognitive impairment in behaving 2VO mice. Modulation of hippocampal low gamma using 30–50 Hz light flicker (1) restored hippocampal CA1 slow gamma and phase-amplitude coupling to theta oscillations in free behaving 2VO mice; (2) protected CA1 neurons; (3) improved locomotion, spatial learning, and memory; and (4) enhanced CA3-CA1 synaptic strength through an RGS12 modulation of N-VGCC-dependent pathway. Future investigations of the precise neural circuitry modulated by the light flicker are warranted.

The current findings that cerebral ischemia affects low gamma in free-behaving mouse and that the 30–50 Hz light flickers can restore low gamma to confer neuroprotection through an RGS12-modulated enhancement of CA3-CA1 synaptic plasticity are novel. Our data that 40 Hz flicker light can modulate hippocampal low gamma are in agreement with published studies[10,12,23,24]. It is worth noting that our studies were performed when the mouse locomotion state was free from generating theta harmonics[20,21]. Flickering light did not cause the mouse to run faster since OFT data shows that these mice had no anxiety behaviors during and after the light treatments (Fig. 3). The PAC strength depends on behavior and is one of the most predictive tools to demonstrate relations to learning and memory[36,37]. The present study revealed theta-low gamma PAC deficits after cerebral ischemia and light flicker restore PAC in behaving mice engaged in an object exploration behavior.

Neuroprotection conferred by 40 Hz light flicker is striking. Strengthening of CA3-CA1 excitatory synapses could send pro-survival signals to the CA1 neurons, making the CA1 neurons

more resistant to reperfusion-induced neuronal death. The proteomic analysis identified groups of genes regulating synaptic plasticity and cell death, which may contribute to CA1 neuronal survival. The literature also supports that activation of synaptic NMDAR promotes postsynaptic neuron survival, while extrasynaptic NMDAR promotes neuronal death[38]. L-VGCC activity transmits calcium-regulated survival signaling to the nucleus that supports learning and memory. Inhibiting L-VGCC and LTP$_{L-VGCC}$ causes delayed death of CA1 neurons in 2VO[18,39]. Thus, 40 Hz light drives low gamma to enhance CA3-CA1 synaptic strength to achieve neuroprotection.

There are three lines of evidence supporting 40 Hz light flicker enhancement of synaptic strength. First, the activity-triggered RGS12 modulation of N-VGCCs-dependent presynaptic release probability, which serves as an essential link between presynaptic membrane depolarization and presynaptic vesicle exocytosis in tuning neurotransmission. PPR and FM1-43 data are reliable readouts of presynaptic N-VGCC-initiated fast release of neurotransmitters[40]. Preventing RGS12 binding to the tyrosine-phosphorylated N-VGCC pore-forming α-subunit using RGS12 decoy peptide and rAAV-RGS12 lent further support of RGS12's novel role in enhancing synaptic strength to 40 Hz light flicker treatment.

Second, light flicker restored both LTP$_{NMDAR}$ and LTP$_{L-VGCC}$ in the 2VO hippocampus. Enhancement of excitatory neurotransmitter release by light flicker activated ionotropic NMDAR and could trigger AMPA-type glutamate receptor trafficking to the dendritic surface in an L-VGCC-dependent manner[41]. L-VGCC tunes the surface expression of key synaptic proteins to enhance synaptic strength. Indeed, theta-burst stimulation induces LTP$_{L-VGCC}$[42]. Inactivation of the Ca$_V$1.2 gene causes deficiencies in NMDAR-independent LTP and impairment in spatial learning[18,32].

Third, the formation of stable dendritic spines also represents signs of enhancement of synaptic strength in response to 40 Hz

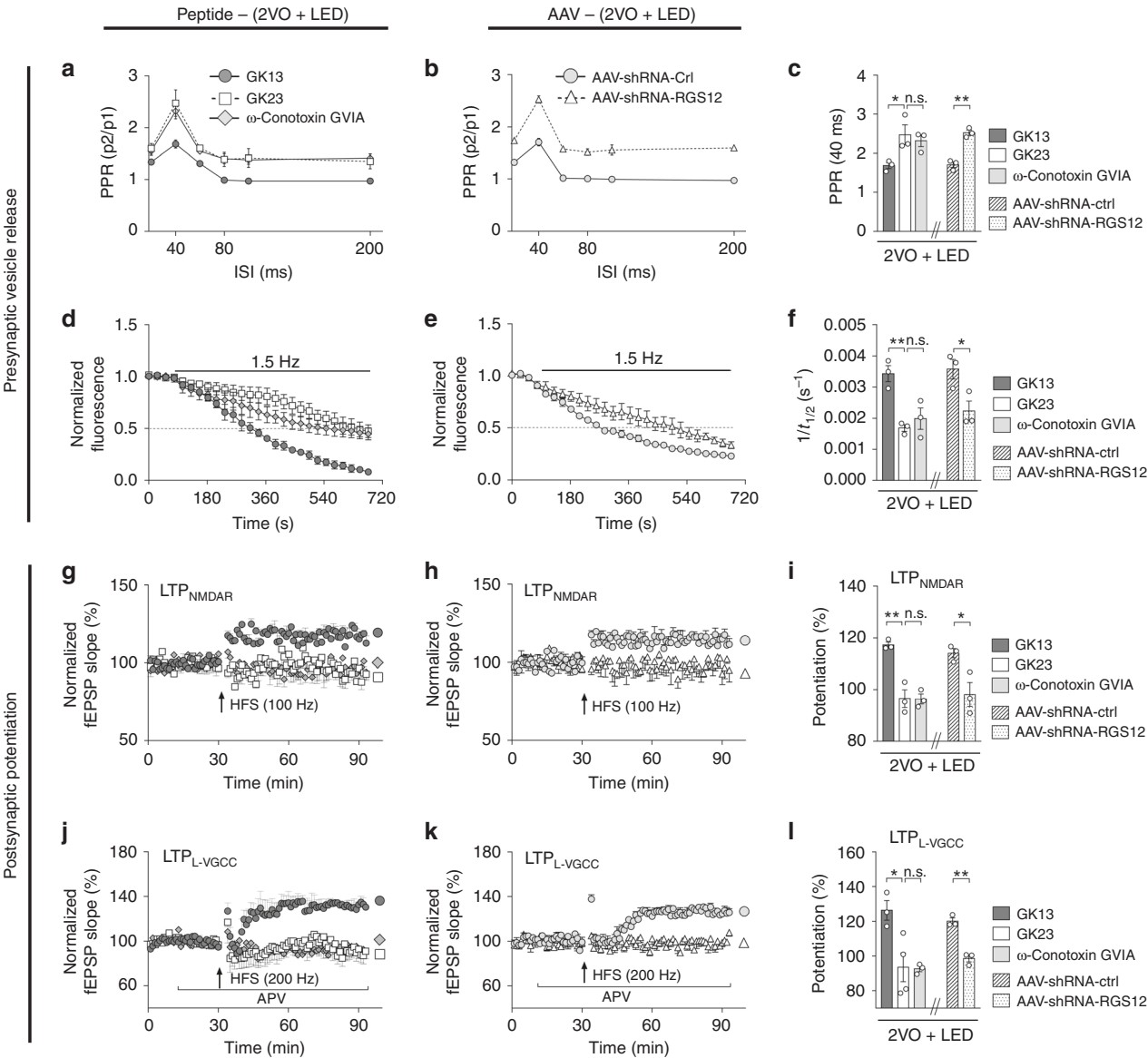

**Fig. 8 RGS12 regulation of N-VGCC-dependent PPR, vesicle release, and LTPs. a** Mouse was implanted with an osmotic pump delivering GK13, GK23, or ω-conotoxin GVIA before the 2VO surgery and 40 Hz light treatment throughout the 3 d periods. Hippocampal slices were subjected to PPR measurements. **b** AAV-shRNA-Ctl and AAV-shRNA-RGS12 were stereotaxic injected into the CA1 area for 3 weeks, followed by 2VO and 40 Hz light treatment (2VO + LED) for 3 d. Hippocampal slices were measured for PPR. PPR at the 40 ms ISIs for both **a** and **b** were measured and plotted in **c**. **a**, **b**, and **c** share the same legends ($n = 3$ mice each group). The hippocampal brain slices of 2VO + LED mice treated either with GK13, GK23, and ω-conotoxin GVIA (**d**) or with AAV-shRNA-Ctl and AAV-shRNA-RGS12 (**e**) were measured for presynaptic vesicle release probability based on the kinetics of FM1-43 release from presynaptic terminals. Rate ($1/t_{1/2}$) of puncta unloading in brain slices was plotted and shown in **f**. **d**, **e**, and **f** share the same legend ($n = 3$ mice each group): **g–l** Measurements of LTPs. The effect of osmotic pump application of RGS12 decoy peptide GK23 (**g**, **i**, **j**, **l**) and AAV-shRNA-RGS12 (**h**, **i**, **k**, **l**) on CA3-CA1 LTPs of 2VO mice 3 d after 40 Hz light treatment (2VO + LED) was determined using 100 Hz (**g**,**h**) or 200 Hz (**j**,**k**) tetanic stimulations (**g–l** $n = 3$ mice in all groups except $n = 4$ mice in $LTP_{L\text{-}VGCC}$ GK 23 group). Data represent the mean ± SEM. Error bars indicate SEM. *$P < 0.05$, **$P < 0.01$. The test used on the left side of **c**, **f**, **i**, **l** 1ANOVA with Dunnett's post hoc test; on the right side of **c**, **f**, **i**, **l** Two-tailed unpaired t-test; **g**, **h**, **j**, **k** RM-2ANOVA with Sidak's post hoc test. Source data are provided as a Source Data file.

light treatment. The shapes of individual dendritic spines are proposed to correlate with their capacity for structural changes. In particular, the stubby and mushroom-shaped boutons located on CA1 apical dendrites were accounted for 90% of the increased boutons. These two types of boutons are spines that endure structural modifications resulting in enhanced LTPs. CA3 spine losses at 7 d and 14 d post-2VO were rescued by light flicker treatment supporting the role of light flicker in the strengthening of CA3-CA1 synaptic connections.

Microglia activation in cerebral ischemia can be either beneficial or detrimental to neurons depending on the timing. Although 40 Hz light activates microglia in Alzheimer's disease mouse brain to clear β-amyloids[10,11], the current data failed to indicate microglia response to light flicker in the ischemia brain. This is based on microglia immunostaining and the lack of microglia-related gene expression in proteomic profiling. The light treatment time may underlie the lack of response of microglia in the current 2VO studies.

## Methods

**Key resources table**. Which contains information about the antibodies used, pharmacological agents, peptides, and viral vectors. The table can be found in the online Source Data file.

**Animals**. Adult male C57Bl/6 mice (3-months-old) were purchased from the Guangdong Medical Laboratory Animal Center. Adult male Thy1-YFP-H mice (3 months old) were purchased from the Jackson Laboratory (USA). Mice were maintained in a pathogen-free SPFII animal facility in a condition-controlled room (23 ± 1 °C, 50 ± 10% humidity). A 12 h light/dark cycle was automatically imposed. Mice were housed in groups of four to five per individually ventilated cages and given access to food and water ad libitum unless otherwise stated in the methods below. Experimenters were blinded to animals' treatments and sample processing throughout the subsequent experimentation and analyses.

**Ethical approval and animal study design**. All animal experiments were conducted according to protocols approved by the Animal Care Committee of the Southern University of Science and Technology (Shenzhen, China). The ARRIVE guideline was followed when designing and performing animal experimentation[43]. Efforts were made to minimize the number of mice used. The inclusion criterion was based on the identical age and sex of the mice. The exclusion criterion was when the mouse failed to survive 2VO surgery at the end of the 14 d periods of the experimentation. To achieve meaningful statistical differences, a minimal of five mice per group were used in the LFP in vivo recordings, behavioral studies, RGS12 gene expression modifications experiments as indicated in the figure legends. For tissue section staining and western blotting experimentation, at least three mice per group were used.

A total of 112 mice with the same age (3-month-old) and sex (male) were first randomly assigned to two groups (56 each) to receive Sham or 2VO surgery, respectively. After the surgery, mice in each group were randomly assigned to receive LED light or normal light treatment. In the end, these mice became four groups to receive Sham, Sham+LED, 2VO, 2VO + LED treatments. Data derived from all qualified animals were included in the analyses and presentation of the results.

**Mouse two-vessel occlusion (2VO) ischemia model**. Bilateral occlusion of the mouse common carotid arteries was performed[44]. Animals were fasted overnight, but free to drink, before surgery. Mice were anesthetized with a mixture of 2.5% isoflurane in $N_2:O_2$ (70:30; flow rate 400 ml/min) in an induction chamber (Product #R540, RWD Life Science, Shenzhen, China). When the animal showed no sign of consciousness, it was moved to the anesthetic face mask supplied with 1.5% isoflurane in $N_2:O_2$ (70:30; flow rate 400 ml/min). The animal's body temperature was maintained at 37.0 ± 0.5 °C using a rectal temperature probe and a heating blanket (Product # TCAT-2 Temperature Controller, Harvard Apparatus, USA). Furs around the ventral neck region were shaved using a hair clipper to expose the skin which was disinfected with iodine followed by 75% ethanol. Under the operating microscope, an incision in the midline of the ventral neck region was made with a scalpel. The superficial fascia was dissected to expose the bilateral common carotid arteries.

Surgery for 2VO should be combined with systemic hypotension (35–40 mm Hg) to induce sufficient brain ischemia[19,44,45]. To do this, the isoflurane concentration was increased to 5% for 2 min, and the carotid arteries were clamped with micro-vessel clamps for 5 min to induce global cerebral ischemia. The isoflurane level was decreased to 0% 2 min before the release of the micro-vessel clamps and resume of blood reperfusion. The neck incision was sutured with 5-0 sterile silk sutures. The animals were moved into a warm (37.0 ± 1 °C) recovery chamber (Product #DW-1, Harvard Apparatus, USA) to prevent post-ischemic hypothermia. For Sham operations, micro-vessel clamping was performed on the bilateral common carotid arteries. The clamping was immediately released to allow for instant reperfusion. The subsequent operation is identical to that performed on animals undergoing cerebral ischemia, including anesthesia processes.

**Measurement of mean arterial blood pressure (MABP)**. The mean arterial blood pressure was measured using a non-invasive blood pressure system (Product #CODA, Kent Scientific Corporation). Before surgery, the occlusion tail-cuff was placed through the tail and towards the base of the tail until encountering resistance. The volume pressure recording (VPR) sensor cuff was then slid up the tail until reaching the occlusion cuff. The MABP recording started 2 min before the onset of administering of 5% isoflurane in the respiratory gases and continued during the 5 min period of 2VO and into the first 2 min of reperfusion. During the 2 min of administering of 5% isoflurane, the MABP dropped to about 40 mmHg, at which time both common carotid arteries were occluded. The MABP stabilized at 35 mmHg during the occlusion period without further manipulation of isoflurane levels. MABP of every mouse undergoing either sham-operation or 2VO surgery was subjected to hypotension and measurement of MABP.

**Measurement of regional cerebral blood flow (rCBF)**. Mouse hippocampal blood flow (hrCBF) was detected by using a laser Doppler flowmetry (moorVMS-LDF2, Moor Instruments, Wilmington, DE) with a 0.5 mm flexible fiber optics. The

longitudinal hippocampal artery provides the blood supply to the ventral and dorsal parts of the hippocampus. It follows a path approximately parallel to the longitudinal axis of the hippocampus and close to the hippocampal fissure. Localization of the longitudinal hippocampal artery was determined using the three-dimensional digital mouse brain atlases[9,46–48]. After shaving the head fur and disinfecting the head skin with iodine and 75% ethanol, an incision was made on the head skin with a scalpel and the tip of the probe was fixed on to the surface of the skull (bregma: −2.0 anterior/ posterior, 1.5 medial/lateral in mm). The intact skull was moisturized by silicone oil to better visualize the vascular landmarks. The hrCBF recording started 2 min before the onset of administering 5% isoflurane in the respiratory gases. Recordings were made during the entire 5 min of occlusion of the common carotid arteries and continued for 2 min into reperfusion. Five mice per group were used to determine 2VO ischemia/reperfusion-induced changes in hrCBF.

Laser speckle flowmetry: Total cortical blood flow was also measured using a laser speckle contrast imager (MOORFLPI-2, Moor Instruments, Wilmington, DE). Four groups of mice were measured after 3 d, 7 d, and 14 d of 2VO. To reduce movement-induced blood flow artifacts, mice were anesthetized during the measurement following the instrument manufacturer's instructions. The skin of the scalp was cut along the sagittal plane to expose the braincase for imaging. A bolus of saline was applied to the skullcap to normalize the effect of light refraction between animals. Blood flow of the region of interest, such as the visual cortex, was selected and measured using the onboard software (mFLPI2MeasV2.0 for recording and moorFLPIReviewV50 for analysis).

**Visual stimulation protocol**. The visual stimulation equipment, TangGuang[TM], consisted of seven modules, including a tunable frequency signal generator and six LED lamps (36 V). The six LED lamps were connected by parallel circuits and equally distributed around the two transparent cages, which separately housing the control and experimental groups, respectively, at the same time in order for the animals to simultaneously receive light stimulation. The first session (1 h) of LED light flicker visual stimulation was performed 2 h after surgery. The stimulation was conducted twice per day with the first session between 8–9 a.m. and the second session between 8–9 p.m. throughout the whole experimental duration (up to 14 d). Delayed light treatment was also performed 3 d after the 2VO and lasted up to 7 d post-ischemia. In total, 10 Hz, 30 Hz, 50 Hz, 80 Hz, and random frequency LED light flicker visual stimulation were conducted on mice using the same protocol. The animal's body temperature was monitored and maintained at 37.0 ± 0.5 °C using a rectal temperature probe during the light treatment.

**Intracranial electrode implantation surgery**. Seven days before 2VO or Sham surgery, mice were prepared for implantation of electrodes to record local field potentials (LFPs). Animals were fasted overnight, but freed to drink, before surgery. Mice were anesthetized with a mixture of 2.5% isoflurane in $N_2:O_2$ (70:30; flow rate 400 ml/min) in an induction chamber. Anesthesia was maintained with 1.5% isoflurane in $N_2:O_2$ mixture during surgery using an isoflurane vaporizer (Product #R540, RWD Life Science, Shenzhen, China). The animal's body temperature was maintained at 37.0 ± 0.5 °C using a rectal temperature probe and a heating pad (Product #TCAT-2DF, Harvard Apparatus, USA).

A cranial window (1.2 mm in diameter, anteroposterior, A/P: 2.0 mm; mediolateral, M/L: 1.5 mm) over the left hippocampus was created using a dental drill (Product #78001, RWD Life Science, Shenzhen, China) guided by stereotaxis (Product # 68861 N, RWD Life Science, Shenzhen, China). A 4-channels microwire array electrode (35 μm, Stablohm 650, California Fine Wire Co., USA) was inserted into the hippocampal stratum radiatum in area CA1 and immobilized using kwik-sil (Item #KWIK-SIL, Microprobes for Life Science, Gaithersburg, MD, USA). The electrode was anchored to the skull bone using 4 skull screws (1.0 mm diameter) and embedded in dental cement. Mice were then moved into a warm (37.0 ± 1 °C) recovery chamber (Product #DW-1, Harvard) for 1 h.

**LFP recording and analysis**. After 4 d recovery from electrode implantation surgery, mice were adapted to recording position by placing them in a box (50 depth X 50 width X 50 length in cm) 10 min per day until the first recording session. Before each recording session, the box was cleaned by 75% ethanol. During recording, the implant was gently held by a helium balloon allowing the mouse to move freely in the box. To avoid the impact of movement speed on the linearity of theta waves[22], three low-speed movement states were used for the study, i.e., approach, exploration, and low-speed free walking. These low-speed states are relevant to 2VO mice since the mouse does not normally engage in fast running after the surgery. The speed/velocity of movement for the three states was at 2~3 cm/s forward, as determined by footfall patterns[22].

Head-stage was connected to the OmniPlex Neural Recording Data Acquisition System (Plexon Inc, Dallas, TX). A camera was set in the front of the box to record the behavioral states of the mouse. An LED lamp was put in front of the box to generate different frequency lights. LFPs in the CA1 region were recorded for at least 20 min for each mouse. Each mouse received the same pattern of frequencies of light stimulation with a 2 h break in between. These light flickers were at 10 Hz, 30 Hz, 40 Hz, 50 Hz, 80 Hz and an arrhythmic frequency. A total of five mice in each group were used for statistical analysis. The LFP signals were sampled at 1000

Hz with a band-pass filter set at 0.5−300 Hz. Raw data were stored for later off-line analysis.

For LFP data analyses, a multitaper fast Fourier transform method was implemented for power spectral analyses using the NeuroExplorer software (Version 5, Nex Technologies, Colorado Springs, CO, USA). Data were filtered from 0 to 120 Hz using the Digital Filter of Continuous Variables function within the NeuroExplorer software. Frequency-Power curve was generated using NeuroExplorer software Power Spectra for Continuous function. Fig. 1h low-speed walking spectrogram was generated using the NeuroExplorer software Spectrograms function. Time-frequency spectrograms were generated using the NeuroExplorer software *Perievent Spectrograms* function. The power spectrum in Fig. 1i,j,k was given by a multi-taper estimation method using MATLAB using Eq. (1). Given a time series $X_n$, $n = 1, 2, …, N$, the number of the Slepian sequences is $K = 2NW−1$. The simplest multi-taper estimate of the spectrum is given by

$$S(f) = \frac{1}{K} \sum_{k=1}^{N} \left| \frac{1}{N} \sum_{n=1}^{N} \exp(2\pi i f n) u_n^k X_n \right|^2 \tag{1}$$

Where $u_n^k$, $n = 1, 2, … , N$ is the $k$th Slepian sequence.

Phase-amplitude coupling analyses of coupled oscillations were obtained by bandpass filtering signals between 4 and 12 Hz with a Hilbert transform[36]. The MATLAB code used can be referenced in previous published works[49]. The phase-amplitude comodulogram scans multiple frequency pairs searching for cross-frequency coupling.

Two marks were manually added to the signal traces to indicate the times of recording during approach and exploration, and a stable free walking period. Only signal segments between these marks were analyzed. Among the overall recording during these periods, 2 s × 10 representative epochs free of major artifacts were randomly selected for calculating LFP power and coherence. Sections of LFP data from at least 2 s of rest period were selected for the performance of event-aligned Time-frequency analysis using *Perievent Spectrograms* function [Frequency Range: 0–120 Hz or 30–50 Hz; Shift Type: Absolute (s); Number of Shifts : 20; Shift (s): 0.1; Tapers : Single; Windowing Function: Hann]. We calculated the ratio of power spectral density between PSD$_{exploration}$ and PSD$_{approach}$ in theta, low gamma, 40 Hz region, and high gamma.

**Mouse behavioral tests**. All behavioral tests were performed between 2 and 6 p.m. The apparatus was carefully cleaned by wiping with 75% ethanol after each experiment to remove the olfactory distraction. The person performing the test was blind to the animal groups. All mice were familiarized with the researchers[50].

**Open field test**. Mouse spontaneous activity in the open field was tested using a clear plastic cube box (41× 41× 38 cm) under a camera. The center region of the box field was defined as a 20 cm × 20 cm virtual area. Each mouse was given 10 min to move freely in the box. All traveled distances and the distance trace center were recorded and analyzed using Panlab software Smart (V3.0, Harvard Apparatus, Holliston, MA). The rearing numbers, representing the numbers of times mouse lifting the forepaws to explore upwards, were recorded by the experimenter.

**Morris water maze (MWM)**. Morris water maze was performed to determine the spatial learning ability and reference memory. MWM was performed in a blue water pool (120 cm in diameter with 2/3 of transparent water) containing a circular bright blue platform (14 cm in diameter and submerged 1.5 cm beneath the water surface). Various patterns were equally distributed around the wall as visual cues for mice during the experiment.

For spatial acquisition test, the spatial acquisition learning ability training consisted of five consecutive days with every training day comprising of four trials with 15 min inter-trial interval. The entry points were randomly selected each time from the different designated locations. Once the mouse successfully found the platform within 60 s, it was placed into a cage under a warming lamp as a reward. Otherwise, the mouse was gently and manually guided to the platform and allowed to remain there for at least 20 s. The escape latency and pathlength to the platform was recorded each day to assess the spatial learning ability.

On day 6, a probe trial test was performed. The hidden platform was removed from the pool, and the mouse was placed in the quadrant diagonally opposite the target quadrant and allowed to swim for 60 s. The percentage of time spent in the target quadrant was recorded and analyzed as a measure of spatial memory retention (or the reference memory). Mice were monitored by a camera, and its trajectory was analyzed using the Smart (V3.0, Panlab Harvard Apparatus).

**Y-maze spontaneous alternations**. To assess the spatial working memory of the mice, spontaneous alternation behavior was assessed in a Y maze after the MWM. The Y maze used in this test was composed of three identical acrylic light blue arms (31 cm long, 17 cm high, and 9.5 cm wide). During each trial, the mouse was placed at the maze center and allowed to explore freely through the maze for 5 min. The sequence of arm entries was recorded by a monitor right above the apparatus and analyzed using the video tracking Panlab software (Smart V3.0, Harvard Apparatus). Alternation percentage (%) was calculated as a proportion of arm choices differing from the previous two choices. To minimize the difference made by

experimenter handling, mice were handled by the same one experimenter throughout all behavioral experiments.

**Acute hippocampal slices**. Mice (either C57BL/6 or Thy1-YFP-H strains) were euthanized by cervical dislocation. Brains were immediately harvested and placed into ice-water cutting solution composed of (in mM): 124 NaCl, 2.5 KCl, 1.25 NaH$_2$PO$_4$, 25 NaHCO$_3$, 2 CaCl$_2$, 2 MgSO$_4$, 15 glucose, and 220 sucrose, saturated with 95% O$_2$/5% CO$_2$ (v/v). Coronal slices (300 μm thickness for electrophysiology recording and 400 μm thickness for two-photon imaging) containing the hippocampus were cut using a vibratome (VT1120S, Leica Systems). Hippocampal slices were rinsed twice in artificial cerebrospinal fluid (aCSF), which composed of (in mM) 124 NaCl, 2.5 KCl, 1.25 NaH$_2$PO$_4$, 25 NaHCO$_3$, 2 CaCl$_2$, 2 MgSO$_4$, and 15 glucose. Slices were gently moved to a brain slice keeper containing aCSF saturated with 95% O$_2$/5% CO$_2$ (v/v). Incubation of slices at temperature 37 °C for 30 min before transferring to room temperature for at least 1 h prior to recording or imaging. Slices were then individually transferred to a recording chamber fixed to the stage of an upright microscope. During recording or imaging, slices were continuously perfused with aCSF saturated with 95% O$_2$/5% CO$_2$ (v/v) at a rate of 3 ml /min.

**Electrophysiology of LTP and paired-pulse ratio (PPR)**. Coronal hippocampal slices (300 μm) were prepared using a Vibratome slicer (VT1120S; Leica) in ice-cold aCSF. All recordings were performed in a submerged recording chamber with aCSF (32 ± 0.5 °C, 3 ml/min), saturated with 95% O$_2$/5% CO$_2$ (v/v)[50,51]. Injection of aCSF into the glass pipettes (1–5 MΩ) allowed the recording of the field excitatory postsynaptic potentials (fEPSPs) evoked in stratum radiatum of CA1 region by stimulating Schaffer collateral/commissural (Sch/com) through a concentric bipolar stimulating electrode. After slice recovery in the recording chamber (10 min), input-output curves relating stimulus current intensity to fEPSP slope and amplitude were generated. The stimulus intensity required for the half-maximal fEPSP slope was selected.

PPR experiments were performed using inter-stimulus intervals (ISIs) of 20, 40, 60, 80, 100, and 200 ms. PPF (%) was estimated by the ratio of the second pulse-induced fEPSP slope to the first one. Baseline responses were recorded per 30 s for 30 min before tetanus stimulation and normalization.

NMDA receptor-dependent LTP (LTP$_{NMDAR}$) was induced by applying a high-frequency stimulation (100 Hz, 1 s). L-type voltage-gated calcium channel-dependent LTP (LTP$_{L-VGCC}$) was induced with four stimulus trains (200 Hz, 0.5 s) delivered at a rate of 1 train per 5 s in the presence of an NMDA receptor antagonist, D-APV (50 μM, SIGMA). An L-VGCC blocker, Nifedipine (10 μM, SIGMA), was applied to reduce the evoked responses induced in the presence of D-APV. For PPF experiments, brain slices were preincubated in D-APV or Nifedipine for 20 min before recording. For LTP recording, D-APV and Nifedipine were delivered after 10 min onset of baseline recordings. ACSF containing D-APV or Nifedipine or both were used to perfused brain slices during recording. Individual brain slices were used to PPF, LTP$_{NMDAR}$, and LTP$_{L-VGCC}$ recoding, respectively.

Both LTP$_{NMDAR}$ and LTP$_{L-VGCC}$ were determined according to responses between 45 and 60 min after conditioning stimulation. The evoked responses were recorded in current-clamp mode using Axon MultiClamp 700B (Molecular Devices, USA) amplifier.

Data were acquired and analyzed with pClamp 9 (Molecular Devices, Sunnyvale, CA) and GraphPad Prizm7.0 (San Diego, CA). Slices from each animal were averaged together to give an n value of one. Statistical analyses of pooled data were performed by Students t-test, one sample t-test vs. a theoretical mean of zero or 100%, or repeated measure two-way ANOVA. $P < 0.05$ was considered to be statistically significant. Data are presented as mean ± SEM with the number of determinations ($n$) representing the number of recordings.

**Two-photon excitation microscopy**. The Thy1-YFP-H transgenic mice selectively express yellow fluorescent protein at high levels in hippocampal pyramidal neurons as well as subsets of central neurons. Axons are brightly fluorescent all the way to the terminals making this type of mice widely used for in vivo imaging studies. An upright FVMPE-RS multiphoton microscopy system (Olympus, Japan) based on an Olympus upright microscope equipped with a water objective (25X/1.05, W.D. = 2 mm) was used. A femtosecond-pulsed Ti:Sa laser (Mai Tai DeepSee, Spectra-physics, USA) was applied to produce 960 nm laser for excitation of YFP protein expressed in hippocampal pyramidal neurons.

Laser power was 120 mW under the objective. Hippocampal slices were continuously perfused at 3 ml / min with aCSF, saturated with 95% O$_2$/5% CO$_2$ (v/v) for imaging. Images of CA1 and CA3 regions were taken with 512 x 512pixel resolutions and 0.99 μm per pixel size. Both apical dendrite and basal dendrite were imaged under the same resolution with 0.1 μm per pixel size. To quantify apical dendritic spines of CA1 and CA3 neurons, Scholl analysis[34] was performed on the second and third-level branches within the region of interest (ROI). At least 6 dendrites in CA3 and CA1 region for both apical and basal dendrite on each hippocampal slice were imaged.

All images were analyzed using ImageJ (NIH) and Imaris (Version 9.2, Oxford Instruments) software. Dendritic spines were classified into a stubby, mushroom,

and thin shapes based on their morphology, and each class was counted separately on a given length of the dendrite. Spine density was averaged per length of dendrite and plotted.

**Visualization of presynaptic vesical release: loading.** Presynaptic boutons were loaded by bath-applying 5 μM FM1-43 (Molecular Probes, Eugene, OR) in 45 mM K$^+$ aCSF for 15 min with D-APV (50 μM) added to the external solution to prevent synaptic driven action potentials from accelerating dye release[33,52]. After dye-loading, slices were transferred to aCSF containing ADVASEP-7 (0.1–0.15 mM) for 30–35 min to remove any dye bound to extracellular tissue. The stimulus-induced distaining was evoked using 1.5 Hz stimulation, as the bipolar tungsten electrode placed into the CA1 stratum radiatum region at a depth of approximately 50–100 μm. During this process, the D-APV was always in the external solution to prevent excitotoxic damage and the induction of plasticity caused by synaptic stimulation.

**Visualization of presynaptic vesical release: imaging.** A Two-photon patch-clamp system (Scientifica, Hyperscope, MMTP-3000, UK; HEKA, EPC-10 USB Double, GER) equipped with a Ti:sapphire laser (900 nm; Coherent, Chameleon Ultra II) was used for two-photon excitation microscopy. Images were acquired using a 16 X 0.9 NA water-immersion ultraviolet objective (CFI75 LWD) with Lasersharp software provided by BioRad. A series of 6 images (512 × 512 pixels, 0.077 μm/pixel in the $x$–$y$ axes) at different focal planes were acquired every 26–27 s with a 1-μm step in the $z$-direction. Then a subset of 3–6 contiguous in the $z$-axis series was selected and aligned with the first z-series by shifting each image in 3 dimensions based on the location of the peak of their cross correlograms with the first z-series. The puncta (4–5 in each slice), which was within 50–60 μm of the stimulating electrode and stimulus-dependent unloading, were selected according to the three criteria: first, a fluorescence intensity more than two standard deviations above the mean background; second, a diameter between 0.3 to 1.8 μm; and third, a roughly circular shape. Fluorescence measurements were analyzed using Image J by spatially averaging signals over a region centered over each of the puncta for each time point during the unloading protocol. The activity-dependent destaining time courses were generated by each punctum subtracting its residual fluorescence intensity (<10% of initial intensity), then normalized to the maximal fluorescence intensity of punctum that was in the unloading procedure. The inverse halftime of decay of intensity during unloading ($1/t_{1/2}$) was then calculated.

**Immunofluorescence staining.** Mice were perfused with 0.01 M PBS, and 4% freshly made paraformaldehyde under deep anesthesia[53]. Coronal brain sections were cut at 10 μm thickness using a cryostat microtome (CM1950, Leica). After post-fixation in 4% paraformaldehyde for 15 min, sections were washed with PBS for 5 min, permeabilized in TBS containing 0.3% Triton X-100 for 20 min, and blocked in TBST (0.1% Tween) containing 5% BSA, 0.3% Triton X-100 for 1 h at room temperature. Sections were incubated with the primary antibody overnight at 4 °C in TBST with 1% BSA, 0.3%Triton X-100. Primary antibodies (anti-Iba1: 1:250, ab178847; anti-GFAP: 1:500, ab7260; anti- Parvalbumin: 1:500, ab11427) were visualized with an appropriate secondary antibody (goat Anti-Rabbit IgG H&L (Alexa Fluor® 488), 1:1000~1500, ab150077) conjugated with Alexa-Fluor 488 (Jackson Immunoresearch) after washing 3 times with TBST. Slides were covered using a coverslip under the Fluoroshield Mounting Medium spiked with DAPI (Abcam; ab104139). Images were acquired using a fluorescence microscope (M2, Zeiss), a confocal microscope (LSM 710; Zeiss), and quantified using Image J.

**Fluoro-Jade B Staining.** Fluoro-Jade B staining was performed to visualize degenerating neurons. Brain tissue sections were cut on a cryostat microtome (CM1950, Leica) at a thickness of 25 μm. Sections were collected in 0.1 M PBS and mounted on a 2% gelatin-treated microscope glass slides. All tissue slices were dried at 50 °C for at least 0.5 h. Tissue slices were first immersed into a solution of 100% EtOH for 5 min, followed by rinsing in 70% ethanol and distilled water for 2 min each session. After incubating in 0.06% potassium permanganate solution for 15 min, slides were washed with distilled water and transferred to a 0.001% solution of Fluoro-Jade B (dissolved in 0.1% acetic acid, Millipore, USA) for 30 min. Finally, the stained slides were washed in distilled water and dehydrated thoroughly in a heated incubator at 50 °C for 5 min. Slides were then cleared in xylene for 2 min and mounted using a coverslip under DPX mounting medium (Thermo Scientific). FJB positive cells were reported as the number of FJB-positive cells per CA1 region.

**Blood vessel quantification.** Mice were anesthetized via i.p. injection of sodium pentobarbital (50 mg/kg) in a vehicle containing 0.9% sodium chloride. Upon sedation, mice were transcardially perfused with 100 μg of FITC-tagged tomato lectin (FL-1171, Vector Laboratories, Burlingame, CA) in 100 μl volume injected directly into the left ventricle of the heart, over a period of approximately 30 s. The heart continued to beat for approximately 1 min following the injection. The animal was then perfused with 20 ml PBS followed by 30 ml 4 % paraformaldehyde in PBS. The brain was dissected, embedded in O.C.T. compound. Brain sections at 20 μm were cut on a cryostat directly onto Superfrost Plus slides. The FITC–lectin signal in the hippocampus and visual cortex was determined using a fluorescent microscope, and the blood vessel density was measured using Image J.

**Proteomics analysis of ischemic mouse hippocampus.** C57BL/6 mice were euthanized by cervical dislocation. The hippocampus was quickly harvested on ice and ground to powder in liquid nitrogen. Four volumes of lysis buffer (8 M urea, 1%. protease inhibitor cocktail) were added to the cell powder, followed by soni-cation three times on ice using a high-intensity ultrasonic processor (Ningbo Scientz Biotechnology Co., Ltd. China). The remaining non-soluble components were removed by centrifugation at $12,000 \times g$ at 4 °C for 10 min. The supernatant was collected, and the protein concentration was determined with the BCA kit according to the manufacturer's instructions.

To digest proteins, the protein solution was reduced with 5 mM dithiothreitol for 30 min at 56 °C and alkylated with 11 mM iodoacetamide for 15 min at room temperature in darkness. The protein sample was then diluted by adding 100 mM Triethylammonium bicarbonate (TEAB) to reduce urea concentration to less than 2 M. Trypsin was added at 1:50 trypsin-to-protein mass ratio for the first digestion overnight and 1:100 trypsin-to-protein mass ratio for a second time for 4 h of digestion.

To conduct LC-MS/MS analysis, the tryptic peptides were dissolved in 0.1% formic acid (solvent A), and directly loaded onto a home-made reversed-phase analytical column (15 cm length, 75 μm i.d.). The column gradient comprised of an increase from 6% to 23% of solvent B (0.1% formic acid in 98% acetonitrile) over 26 min, 23% to 35% in 8 min and to 80% in 3 min then holding at 80% for the last 3 min, all at a constant flow rate of 400 nL/min on an EASY-nLC 1000 UPLC system.

The peptides were subjected to NSI source followed by tandem mass spectrometry (MS/MS) in Q Exactive™ Plus (Thermo) coupled online to the UPLC. The electrospray voltage applied was 2.0 kV. The m/z scan range was 350–1800 for a full scan, and intact peptides were detected in the Orbitrap at a resolution of 70,000. Peptides were then selected for MS/MS using the NCE setting as 28, and the fragments were detected in the Orbitrap at a resolution of 17,500. A data-dependent procedure that alternated between one MS scan followed by 20 MS/MS scans with 15.0 s dynamic exclusion was performed. Automatic gain control (AGC) was set at 5E4. The fixed first mass was set as 100 m/z.

To identify proteins, the resulting MS/MS data were processed using the Maxquant search engine (Version 1.5.2.8, Max Planck Institute of Biochemistry). Tandem mass spectra were searched against the E. coli SwissProt database concatenated with reverse decoy database. Trypsin/P was specified as a cleavage enzyme allowing up to 2 missing cleavages. The mass tolerance for precursor ions was set as 20 ppm in the First search and 5 ppm in the Main search, and the mass tolerance for fragment ions was set as 0.02 Da. Carbamidomethyl on Cys was specified as a fixed modification, and oxidation on Met was specified as variable modifications. False discovery rate was adjusted to < 1%, and the minimum score for peptides was set > 40.

Proteins with >1.2-fold of change were selected based on the result of LC-MS/MS analysis for clustering analysis and to generate a heat map. The content of the filtered proteins matrix was transformed by the function $x = -\log10$ (content). The $x$ values were z-transformed. One-way hierarchical clustering (Euclidean distance, average linkage clustering) was employed to cluster z-scores in genesis. Cluster membership was visualized by MeV software (Version 4.90, http://mev.tm4.org/).

Gene Ontology (GO) annotation of proteome was derived from the UniProt-GOA database (http://www.ebi.ac.uk/GOA/). Identified protein IDs were changed to UniProt IDs and matched to GO IDs by protein IDs. Based on GO IDs, retrieved the corresponding information from the UniProt-GOA database. The InterProScan software (Version 5.14–53.0, EMBL-EBI) was used to annotate protein's GO function according to the protein sequence alignment method. If the identified proteins were absent in the UniProt-GOA database, then the proteins were classified by Gene Ontology annotation based on cellular function.

**RGS12 decoy peptides.** The phosphotyrosine binding (PTB) domain at the N-terminus of RGS12 interacts with the SNARE-binding or "synprint" region of N-VGCC in a phosphotyrosine-dependent manner[54]. GK23 amino acid sequence consisting of "RASCEALY(P)NE" was linked with a TAT leader sequence (GRKKRRQRRRPQ) at the N-terminus and a FITC tag added to the C-terminus for easy visualization. The chimeric polypeptide was chemically synthesized commercially by GL Biochem Ltd. (Shanghai). The sequence is TAT-RGS12-PTB (GK23) = GRKKRRQRRRPQ-Arg-Ala-Ser-Cys-Glu-Ala-Leu-Tyr(P)-Asn- Glu-Lys-FITC; The GK13 control peptide sequence is TAT-FITC(GK13) = GRKKRRQRRRPQ -Lys-FITC. Hippocampal slices were pre-incubated in aCSF containing 250 nM TAT peptides for 1 h at room temperature. Slice was then continuously perfused at 3 ml /min with aCSF saturated with 95% $O_2$/5% $CO_2$ (v/v) in the absence of TAT peptide during electrophysiological recording. The penetration of the TAT peptide into the hippocampal tissue was confirmed by examining the slices under a fluorescent miscopy to detect peptide-linked FITC. The FITC fluorescence remained in the slices after 1.5 h of electrophysiological recording.

**Intracerebroventricular administration of peptides.** GK23 (2 mg/kg/day), GK13 (2 mg/ kg / day), ω-Conotoxin GVIA (2.28 ng/kg/day) were dissolved in aCSF and infused into osmotic pumps (#1003D, Alzet, CA), respectively. The pumps were filled, connected to a cannula (Brain infusion kit 3, Alzet, CA), and incubated in sterile saline (0.9 %) at 37 °C overnight.

Mice were fasted overnight but freed to drink before the surgery. The mouse was fixed to a stereotaxic apparatus (Product # 68861 N, RWD Life Science,

Shenzhen, China), and the midsagittal incision was made by a scalpel. After cleaning the periosteum, an aperture (0.5 mm in diameter) was drilled by a dental drill (Product #78001, RWD Life Science, Shenzhen, China) following the coordinates from bregma: anterior-posterior = −0.5 mm; lateral = 1.0 mm. The cannula was inserted into the hole (depth = 2.5 mm from flat skull surface) by a cannula holder and anchored with dental cement. The osmotic pump was placed into the subcutaneous pocket on the back of the mouse and closed the incision with surgical sutures. After osmotic pump implantation, the 2VO surgery was performed. Mice were then moved into a warm (37 ± 1 °C) recovery chamber (Product #DW-1, Harvard) for 1 h. Visual stimulation was performed 2 h after the 2VO surgery. Peptide infusion and light stimulation lasted for 3 days. Mice were then sacrificed for brain slicing and electrophysiology recordings.

**Stereotaxic injection.** Recombinant Adeno-Associated Viral Vectors (rAAVs, $1.0 \times 10^{12}$ viral particles/ml) were custom made by Hanbio Biotechnology Co. Ltd (Shanghai, China). rAAVs (2 ul). rAAVs were bilaterally injected into the dorsal CA1 area (anteroposterior, A/P: 2.0 mm; mediolateral, M/L: 1.50 mm; dorsoventral, D/V: 1.60 mm) at a rate of 0.2 μl/min using a pump (Legato185, KD Scientific). Mice were allowed to recover for 3 weeks before electrophysiological recordings and visualization of presynaptic vesicle release.

**Western blotting.** The hippocampus was removed and lysed in 200 ml lysis buffer (Product #C500008, Sangon Biotech, China) containing proteinase inhibitor and phosphatase inhibitor. The lysates were centrifuged at 13,680 g for 10 min at 4 °C, and the supernatant was collected and boiled in boiling water for 10 min with loading buffer (4:1 ratio). Whole proteins were electrophoresed in a 10% SDS-PAGE gel and then transferred to polyvinylidene fluoride (PVDF) membranes (0.45 μm) using iBlot 2 dry blotting system (Thermo Fisher Scientific Systems, USA). After blocking in 5% skim milk, the PVDF membranes were incubated with the primary antibody (anti-RGS12: 1:500, sc-514173) overnight at 4 °C. The PVDF membranes were then incubated with a secondary antibody after washing three times with TBST. Protein band intensities were detected with Tanon (Shanghai, China). Equal protein loading was ensured using an internal control β-actin detected by a mouse monoclonal antibody at 1:2000 (#ab6276, Abcam) dilution. Every brain sample was repeated twice independently.

**Statistical analysis.** All analyses were performed using GraphPad Prizm (Version 7.0, San Diego, CA) or SPSS software (Version 22.0, IBM, USA). All data were presented as mean ± standard error of the mean (SEM). The 2VO surgery and LED treatment were used as two main factors for the 2ANOVA analysis. Multiple group comparisons using a 1ANOVA and 2ANOVA were followed by a post hoc Fisher's LSD, Tukey's, Sidak's or Dunnett's test to identify significant groups as indicated in the figure legends. A 2ANOVA with repeated measures over time was applied to the analysis of LFPs, LTP recordings, distance traveled to the platform, and escape latency in the spatial acquisition test of MWM to detect significant differences between groups on different days or treatments. Statistical details for specific experiments - including exact n values and what n represents, precision measures, statistical tests used, and definitions of significance - can be found in figure legends. Where representative images were shown, each experiment was repeated independently at least three times with similar results. A p value < 0.05 was taken to indicate statistical significance. 1ANOVA: one-way ANOVA; 2ANOVA: two-way ANOVA; RM-2ANOVA: repeated measure two-way ANOVA. *$p < 0.05$; **$p < 0.01$; ***$p < 0.001$, ****$p < 0.0001$.

**Reporting summary.** Further information on research design is available in the Nature Research Reporting Summary linked to this article.

## Data availability

The data supporting the findings of this study are available with the article and its Supplementary Information file or are available from the corresponding author upon request. The source data underlying Figs. 1b–g, i–m, o–r; 2c–e; 3b–h; 4a–b, d–g; 5a, b, d–e; 6e–l; 7b–d; 8; Supplementary Figs. 1b, c, f, g, i–n; Supplementary 2b, c, e–h; 4; 5; 6, and 8d–h are provided as a Source Data file. The moue brain atlas images on the left of Supplementary Fig. 2a was derived from Allen Mouse Brain Atlas (http://atlas.brain-map.org/atlas?atlas=1&plate=100960240). Supplementary Fig. 8a protein sequence alignment was performed using CLUSTALW multiple sequence alignment at https://www.genome.jp/tools-bin/clustalw. Source data are provided with this paper.

## Code availability

All the codes used in the current study are available from the corresponding author upon request. Source data are provided with this paper.

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

## Acknowledgements
We thank SUSTech Animal Facility for the maintenance of mice and the SUSTech Core Facility for supports on the use of two-photon microscopy. We also thank Prof Li-Huei Tsai for the encouragement and constructive discussions. Zhenyu Zhang, Yiqi Zhang, Chao Wang, Qiyuan Liang, Shibo Bai, Qiangrui Dong, and Tengxiao Si provided technical assistance. Financial supports to Dr. Sheng-Tao Hou were from grants from the National Natural Science Foundation of China (81571287, 81871026); Shenzhen-Hong Kong Institute of Brain Science-Shenzhen Fundamental Research Institutions (2019SHIBS0002); Shenzhen Science and Technology Innovation Committee Research Grants (JCYJ20140417105742709; JCYJ20160301112230218; JCYJ20180504165806229; KQJSCX20180322151111754); Natural Science Foundation of Guangdong Province (2017A030313808); Shenzhen Sanming Project (Baoan Hospital, 2018.06), State Key Laboratory of Neuroscience Open Competition Grants (SKLN-201403), SUSTech Peacock Program Start-up Fund (22/Y01226109) and SUSTech Brain Research Centre Fund. S.T.H. is also supported by the Guangdong Innovation Platform of Translational Research for Cerebrovascular Diseases and SUSTech-UQ Joint Center for Neuroscience and Neural Engineering (CNNE).

## Author contributions
Z.L. performed animal surgery, electrophysiology for LFP and LTP recordings, mouse behavioral studies, and data analysis. M.Y. performed mouse behavioral studies, immunohistochemistry, western blotting, electrophysiology, visualization of presynaptic vesicle release experiments, and data analysis. R.L. participated in mouse breeding, animal models, LED light treatment experiments, and mouse behavior studies. Y.W. performed genotyping, behavioral studies, two-photon imaging. N.C. performed MATLAB analysis of LFP power spectra. M.W. was responsible for immunohistochemistry. Z.Z. was involved in the initial planning of experiments, and Y.T. and L.W. were responsible for recordings and analysis of LFP in free-moving mice. S.T.H. conceived the idea, designed the experiments, analyzed the results, revised the first draft, and wrote the manuscript. S.T.H. provided financial support to the project through grant funding.

## Competing interests
The authors declare no conflict of interest.
