## [Peer Review File · Nature Communications]

Reviewers' comments:

Reviewer #1 (Remarks to the Author):

This study aims to determine the main mechanism underlying the benefit of 40 Hz flickering light in promoting neuroprotection of CA1 neurons, synaptic plasticity and cognitive function following cerebral ischemia using the 2-vessel occlusion (2VO) mouse model of global cerebral ischemia in susceptible mouse strain C57BL/6 with low PCom patency.

In brief, the investigators applied visual stimulation to mice with rhythmic LED light in 40Hz frequency, which entrained low gamma oscillation (30-50Hz) in the CA1. Light treatment was provided 2 sessions/day, one hour per session, commenced at 2 hrs after 2VO and lasted for up to 14 days for the longest survival group. The initial observation was that it restored CA1 low gamma power in exploring mice with 2VO, reduced CA1 cell death (assessed by FJB staining), improved locomotion, reduced anxiety (open field test) and improved memory performance (water maze test) compared to the 2VO group without light stimulation. The effect on low gamma entrainment was specific, since normal light or random frequency LED didn't have an effect in freely behaving mice. 40Hz LED light didn't affect high gamma in exploration. For the neuroprotection study, they found light frequencies of 30 and 50 Hz were also effective, but not 10, 80 Hz or random frequency. If light treatment was delayed till 3 ds after 2VO, there was much more cell death. Microglia were reduced in the CA1 and DG of the light treated 2VO group, but GFAP remained unchanged compared to the non-treated 2VO group.

In order to investigate the physiological mechanism of this light stimulation on the hippocampus, synaptic plasticity through fEPSP was determined in acute hippocampal slices. Both postsynaptic synaptic plasticity LTPNMDA and LTPL-VGCC was reduced by 2VO but restored by light treatment. Data for all synaptic experiment were presented from the 3-day time point (after 2VO) in the main figures. Parallel experiments provided evidence that 40-HzLED light also improved presynaptic short-term plasticity with data in presynaptic vesicles readily release probability (PPR) determined with paired-pulse ratio at 40 ms and the kinetics of presynaptic vesicles release with fluorescent tracer FM1-43 by multi-photon microscopy. They also provided additional structural evidence showing a reduction of spines in apical and basal dendrites in CA1, but not in CA3, by 2VO and restored by light treatment.

Then proteomic analysis was performed on the whole hippocampus to search for candidate molecular targets. The authors decided to focus on RGS12, a G protein that regulates the N-type voltage gated calcium channel for neurotransmitter release, of which the level of expression was reduced by 2VO but restored by light treatment. By using a synthetic decoy peptide that blocks the interaction between RGS12 and N-VGCC via i.c.v delivery, the authors were able to demonstrate an abolishment of the effect of 40Hz light on presynaptic vesicles release and LTP. Similarly annulment was observed with knock down of RGS12 with an AAV siRNA vector. This is clearly the crescendo of the study. Both experiments were well controlled. The AAV knock down also reduced low gamma but not high gamma power in freely exploring mice. With the converging evidence, the authors claimed that the 40HzLED light improved synaptic plasticity and cognition via RGS12 by positively modulating N-VGCC. The enhanced CA3-CA1 synaptic strength then conferred neuroprotection.

Recently there have been a series exciting studies demonstrating the power of neuromodulation in reducing Alzheimer pathology with either 40Hz light or sound, targeting brain low gamma oscillation. Thus, this study is very timely. Although some mechanisms have been elucidated in earlier reports, the involvement of RGS12 acting on N-VGCC from this paper is clearly very novel. Besides high quality data centering on synaptic plasticity as the core, this study also employs multidisciplinary approaches, which is the main strength. The progression of presentation was especially logical. The authors have shown it for the first time that a noninvasive therapy with 40hz LED stimulation could not only mitigate CA1 neuronal death but also improves synaptic

plasticity and the recovery of cognition. Overall, this is a very well written manuscript with clear, strong and mostly consistent and well-presented data that support their claim. The following comments and suggestions are for the authors to consider.

1. 2VO is essentially a model of global cerebral ischemia, which occurs as a result of cardiac arrest, not stroke. There is great distinction in pharmacology as well as pathophysiology between the two. In order to avoid confusion, I suggest using "cerebral ischemia" instead of "stroke" in describing your results.
2. Global ischemia also induces cell death in other brain regions including the cortex and lateral striatum. Within the hippocampus, somatostatin (+) neurons are said to be more vulnerable than the CA1 neurons to global ischemic insult. It would be useful to characterize your model by showing areas of cell death and determine whether other regions also benefited from the 40Hz light therapy. If so, the mechanism of protection might extend beyond CA3-CA1 synaptic plasticity.
3. Along the same vein, it would be interesting to determine whether light therapy improves blood flow in visual cortex and the hippocampus. More sensitive equipment like laser speckle imaging might be required to detect changes. If possible, [¹⁴C] iodoantipyrine for postmortem analysis would work too. Also, particularly in the 7d and 14d time points, vessel density in the hippocampus should be determined in the 2VO with and without 40Hz light therapy.
4. Water maze acquisition should include pathlength data.
5. It's unclear why the proteomics was conducted in 7d time point while the majority of the data were based on the 3d time point. From the perspective of neuroprotection, the protective mechanism should be activated very early on in order to prevent the delayed neuronal death in the CA1.
6. The authors once emphasized that the 2VO didn't affect CA3, a needed feature for their experiment. However, in figure S6, it appeared the spine density in CA3 began to deteriorate after d3 in the 2VO group. Can the authors elaborate what effect this might have on the synaptic plasticity?
7. Did blocking RGS12 also reduce neuroprotection in the CA1 in the light treated 2VO group?
8. Figure S7, the authors should list the top 5 or 10 up- and down-regulated proteins. Simply describing the numbers is hard to get a sense about the extent of changes upon each treatment.

Jialing Liu

Reviewer #2 (Remarks to the Author):

The major claims of this paper are that 40 Hz light stimulation induces gamma rhythms in the hippocampus, improves hippocampal-dependent memory, improves synaptic plasticity, and is neuroprotective in a mouse model of stroke. The paper includes a variety of different physiological, behavioral, and anatomical methods, but many of the tests were not done rigorously. The conclusions of the paper are not convincingly supported by the results presented. The hypotheses and rationale for performing many of the experiments are also often unclear. Specific comments are detailed below:

Major Points:

1. The authors do not convincingly demonstrate that the 40 Hz LED stimulation generates 40 Hz oscillations in the hippocampus. Specific points related to this general point are listed below.
 - a) In Figure 1C, it appears as though 1/f noise differs between normal light and 40 Hz LED groups. Also, the 40 Hz LED stimulation is associated with a sharp peak at 40 Hz, which is not physiological and is instead likely to be a stimulation artifact.
 - b) The recordings shown in Figure 1B are too low resolution to see anything.
 - c) Why are there 2 peaks in Figure 1B (below)? Could these be theta harmonics? What was observed at lower frequencies?
 - d) Hippocampal rhythms are affected by animals' behavior (e.g., movement speed). There is no

demonstration that behavior was the same across groups. For example, it is possible that the effects were observed because the light stimulation made the mice run faster.

e) There are no peaks observed in Figure 1E, just broadband power. It is unclear what is actually being measured here when power is taken from the 30-50 Hz band. Were any differences observed at lower frequencies?

f) The animals' movement speed likely differed during novel object exploration, but the authors did not appear to control for differences in movement speed.

2. Another major problem with the paper is that the authors conclude that 40 Hz stimulation produces the effects that they observe, but, at least for most of the figures, there were no controls performed using other types of sensory stimulation. So, it is unclear whether any type of sensory stimulation, not just 40 Hz light flickering, would produce similar effects as those reported in Figures 4-7.

3. Does this light stimulation produce temperature differences that could affect recovery from stroke? One could imagine that those LEDs could produce some heat.

4. There is no interaction effect reported to support the authors' claim that slow gamma is the hippocampal rhythm that is selectively affected in 2VO mice.

5. Figure 3B-C does not seem to control for potential differences in running speeds across groups. A similar point applies to Figure 3E. Could this again be due to differences in movement speed?

6. Regarding statistical analyses, why was a repeated measures analysis not performed for the LTP measurements?

7. The authors make statements that are not supported up by the results, even if the results were convincingly supported (e.g., "Enhancing CA1 low gamma power using 40 Hz light alleviated 2VO-induced cognitive deficits through the strengthening of CA3-CA1 synapses").

8. The explanation for the different types of light stimulation is unclear. Were the different stimulation patterns (e.g., 10 Hz, 30 Hz, etc.) delivered in different mice?

9. It is unclear exactly how the novel object exploration fit in. Were all mice subjected to this paradigm? Or only those mice implanted with electrodes? On which days exactly did this occur?

Minor Points:

1. The rationale for looking at NMDA-R-independent LTP is unclear, considering that the authors test memory in the Morris water, and this type of memory has been shown to require NMDA receptors (a Richard Morris Nature paper from the 1980's).

2. The rationale for the open field test was unclear. Why would the 2VO manipulation be expected to affect anxiety or behavior on this test?

Reviewer #3 (Remarks to the Author):

This is an interesting study which aims to determine if entrainment of hippocampal gamma could alleviate stroke-induced cognitive dysfunction. The authors present the case that post-stroke motor and cognitive dysfunctions are associated with the perturbation of specific brain oscillation frequencies and, importantly, such oscillations have the potential to be non-invasively and externally modulated. This is a comprehensive study which has the potential to provide novel and interesting data to the field. However, some methodological concerns need addressing;

The authors have performed two-vessel occlusion of the common carotid arteries which produces a selective damage of the CA1 pyramidal neurones – the authors need to be clear what (disease) they are modelling – 2VO offers a global and transient ischemia that more clinically represents cardiac arrest rather than the regional focal ischemia seen following cerebral stroke which is achieved via other experimental methods. I think it is rather an over extrapolation to suggest that results obtained from this study may therefore have clinical relevance as a therapeutic for stroke prior to such work being conducted in a regionally focal model of stroke.

The justification of the inclusion of sham animals is unclear given the statistical comparison is 2VO vs 2VO+LED.

It's unclear why only 5 animals were used to determine ischemia/reperfusion-induced changes in rCBF? Were the same 5 animals used to determine the changes in MABP?

In vivo experiments should be reported according to the Arrive guidelines and therefore include details of; total animal used, inclusion/exclusion criteria, how animal numbers were determined, if any, and for what reasons, animals were excluded. In some instances (e.g. Morris water maze) n values (4) appear low – how do the authors know the appropriate number of animals were used? Presentation of results is a little confusing as does not match order of experimental detail provided in methods. For example, I would have expected reporting of rCBF and MABP data first. Also it would be useful for the authors to show confirmation of CA1-selective damage following the 2-VO. The authors need to show that the two x 2VO experimental groups received the same amount of damage (i.e. change in rCBF) and that the only difference between the groups was the presence, or absence, of LED.

Reviewer #1 (Remarks to the Author):

This study aims to determine the main mechanism underlying the benefit of 40 Hz flickering light in promoting neuroprotection of CA1 neurons, synaptic plasticity and cognitive function following cerebral ischemia using the 2-vessel occlusion (2VO) mouse model of global cerebral ischemia in susceptible mouse strain C57BL/6 with low PCom patency.

In brief, the investigators applied visual stimulation to mice with rhythmic LED light in 40Hz frequency, which entrained low gamma oscillation (30-50Hz) in the CA1. Light treatment was provided 2 sessions/day, one hour per session, commenced at 2 hrs after 2VO and lasted for up to 14 days for the longest survival group. The initial observation was that it restored CA1 low gamma power in exploring mice with 2VO, reduced CA1 cell death (assessed by FJB staining), improved locomotion, reduced anxiety (open field test) and improved memory performance (water maze test) compared to the 2VO group without light stimulation. The effect on low gamma entrainment was specific, since normal light or random frequency LED didn't have an effect in freely behaving mice. 40Hz LED light didn't affect high gamma in exploration. For the neuroprotection study, they found light frequencies of 30 and 50 Hz were also effective, but not 10, 80 Hz or random frequency. If light treatment was delayed till 3 ds after 2VO, there was much more cell death. Microglia were reduced in the CA1 and DG of the light treated 2VO group, but GFAP remained unchanged compared to the non-treated 2VO group.

In order to investigate the physiological mechanism of this light stimulation on the hippocampus, synaptic plasticity through fEPSP was determined in acute hippocampal slices. Both postsynaptic synaptic plasticity LTPNMDA and LTPL-VGCC was reduced by 2VO but restored by light treatment. Data for all synaptic experiment were presented from the 3-day time point (after 2VO) in the main figures. Parallel experiments provided evidence that 40-HzLED light also improved presynaptic short-term plasticity with data in presynaptic vesicles readily release probability (PPR) determined with paired-pulse ratio at 40 ms and the kinetics of presynaptic vesicles release with fluorescent tracer FM1-43 by multi-photon microscopy. They also provided additional structural evidence showing a reduction of spines in apical and basal dendrites in CA1, but not in CA3, by 2VO and restored by light treatment.

Then proteomic analysis was performed on the whole hippocampus to search for candidate molecular targets. The authors decided to focus on RGS12, a G protein that regulates the N-type voltage gated calcium channel for neurotransmitter release, of which the level of expression was reduced by 2VO but restored by light treatment. By using a synthetic decoy peptide that blocks the interaction between RGS12 and N-VGCC via i.c.v delivery, the authors were

able to demonstrate an abolishment of the effect of 40Hz light on presynaptic vesicles release and LTP. Similarly annulment was observed with knock down of RGS12 with an AAV siRNA vector. This is clearly the crescendo of the study. Both experiments were well controlled. The AAV knock down also reduced low gamma but not high gamma power in freely exploring mice. With the converging evidence, the authors claimed that the 40HzLED light improved synaptic plasticity and cognition via RGS12 by positively modulating N-VGCC. The enhanced CA3-CA1 synaptic strength then conferred neuroprotection.

Recently there have been a series exciting studies demonstrating the power of neuromodulation in reducing Alzheimer pathology with either 40Hz light or sound, targeting brain low gamma oscillation. Thus, this study is very timely. Although some mechanisms have been elucidated in earlier reports, the involvement of RGS12 acting on N-VGCC from this paper is clearly very novel. Besides high quality data centering on synaptic plasticity as the core, this study also employs multidisciplinary approaches, which is the main strength. The progression of presentation was especially logical. The authors have shown it for the first time that a noninvasive therapy with 40hz LED stimulation could not only mitigate CA1 neuronal death but also improves synaptic plasticity and the recovery of cognition. Overall, this is a very well written manuscript with clear, strong and mostly consistent and well-presented data that support their claim. The following comments and suggestions are for the authors to consider.

A: We really appreciated Reviewer #1's encouraging comments. We would like to thank this reviewer for the time spent to provide a thorough review of this manuscript which definitely improved its quality. We have now provided all the required experiments and the manuscript is revised accordingly. New data lent further support to our original conclusions.

Q1. 2VO is essentially a model of global cerebral ischemia, which occurs as a result of cardiac arrest, not stroke. There is great distinction in pharmacology as well as pathophysiology between the two. In order to avoid confusion, I suggest using "cerebral ischemia" instead of "stroke" in describing your results.

A1: We thank the Reviewer #1 for pointing this out. Reviewer #1 is correct and the offensive word "stroke" has been replaced with "cerebral ischemia" in the revised manuscript.

Q2. Global ischemia also induces cell death in other brain regions including the cortex and lateral striatum. Within the hippocampus, somatostatin (+) neurons are said to be more vulnerable than the CA1 neurons to global ischemic insult. It would be useful to characterize your model by showing areas of cell death and determine whether other regions also benefited from the 40Hz light therapy.

If so, the mechanism of protection might extend beyond CA3-CA1 synaptic plasticity.

A2: We thank Reviewer #1 for the suggestions and agree that it is important to examine other areas of the brain to determine the vulnerability of neurons in our 2VO model and whether 40 Hz light protects them. Because the purpose of the present study was to use CA1 selective cell death to examine CA3-CA1 network response to 40 Hz light, our 2VO model was empirically established to producing selective CA1 damage. This is achieved by monitoring hippocampal regional blood flow using laser Doppler flowmetry in combination with inducing systemic hypotension at 35-40 mmHg as shown by the mean arterial blood pressure during 2VO. The occlusion time was set for only 5 min. This model, therefore, produced only selective CA1 cell death and a very small amount of cell death in the DG area. This occlusion time-dependent brain damage following 2VO is supported by previous publications (Kristian and Hu, 2013; Onken et al., 2012, all cited). Importantly, because the damage to the DG region is so minor and varies from mouse to mouse after 2VO, it not possible to quantify whether the light treatment has a protective effect on DG. The characterization of the model and citation to the published references are added in the revised MS pp. 8-9, Fig S2a-c.

Immunohistochemical staining for inhibitory interneurons was performed using an antibody to Parvalbumin. The quantification results showed that PV+ neurons in the hippocampus did not change significantly between the 2VO and Sham groups. Most interestingly, 40 Hz light treatment appeared to have increased the numbers of PV+ interneurons in the hippocampus of 2VO and Sham groups, albeit the lack of a statistical significance. Future investigation is warranted. We added these new data on pp. 8-9 and in Fig S2a-c to show the lack of differences in PV+ cells in the hippocampus between groups.

Q3. Along the same vein, it would be interesting to determine whether light therapy improves blood flow in visual cortex and the hippocampus. More sensitive equipment like laser speckle imaging might be required to detect changes. If possible, [¹⁴C] iodoantipyrine for postmortem analysis would work too. Also, particularly in the 7d and 14d time points, vessel density in the hippocampus should be determined in the 2VO with and without 40Hz light therapy.

A3: We thank Reviewer #1 for the excellent suggestions and agree that these are important issues needing to be fully addressed. To this end, in the revised MS, we have performed new experiments to demonstrate the regional blood flow in the hippocampus of Sham, Sham+LED, 2VO and 2VO+LED groups during surgery using laser Doppler flowmetry. The changes in total cerebral blood flow in general, and CBF in the visual cortex area in particular, were also

measured at 3d, 7d, and 14d after 2VO using laser speckle imaging. At the end of 14d 2VO and light treatment, brains were subjected to perfusion with FITC-Lectin and sectioning to show quantitative blood vessel density in the hippocampus and visual cortex. Indeed, no clear differences occurred between the 2VO and 2VO+LED groups, indicating that light treatment did not significantly affect cerebral blood flow, adding additional supports to our conclusions. The new data and additions are shown on pp 9 and Fig S1b,c, and e-i.

Q4. Water maze acquisition should include pathlength data.

A4: Thank you for pointing this out. The pathlength data are now added in the revised Fig 3f., which showed significant improvement after the light treatment of 2VO mice, confirming our original latency data (Fig 3E). We also added the measurements for speed in the revised Fig 3g which were not significantly different amongst the four groups.

Q5. It's unclear why the proteomics was conducted in a 7d time point while the majority of the data were based on the 3d time point. From the perspective of neuroprotection, the protective mechanism should be activated very early on in order to prevent the delayed neuronal death in the CA1.

A5: We agree with Reviewer #1 that protective genes are activated early on. We have now added the 3d proteomics data. Importantly, the expression of RGS12 was examined using Western blotting on both 3d and 7d groups to experimentally confirm that the levels of RGS12 expression were down-regulated in the 2VO group and up-regulated in the 2VO+LED group, as shown in the revised Fig 7c-e.

Q6. The authors once emphasized that the 2VO didn't affect CA3, a needed feature for their experiment. However, in figure S6, it appeared the spine density in CA3 began to deteriorate after d3 in the 2VO group. Can the authors elaborate what effect this might have on the synaptic plasticity?

A6: We thank the reviewer for pointing this out and apologizes for the confusion. Data in Fig 6 showed changes in CA1 and CA3 dendritic spines at **3d** post 2VO, while Fig S6 showed the spine changes at **3d, 7d, and 14d** after 2VO.

Reviewer #1 is correct that CA3 spine density began to deteriorate at 7d after 2VO and continued into day 14 without light treatment. This may be the result of the loss of connectivity with CA1 apical dendrites. It is important to note that at the **3 d** post-2VO time point, CA3 neurons were intact with no significant loss of dendritic spines (Fig 6f, h, j, i), while, in contrast, CA1 had significant spine loss (Fig 6e, g, I, k, and Fig S6a, b), indicating an early selective damage

to CA1 neurons, but not to CA3 neurons. In the absence of protection to CA1, CA3 spine loss ensues. However, it is interesting that the delayed loss of CA3 spines did not translate into FJB positivity of CA3 neurons at least within the 14 d after 2VO.

These results indicated that early treatment with 40 Hz flicker protected CA1 neurons, which potentially leads to the strengthening of CA1-CA3 synaptic connections and subsequent protection of CA3 spines seen at a later time point of 7d and 14d after 2VO. We have added the detailed explanations to this in the revised MS on pp14.

Q7. Did blocking RGS12 also reduce neuroprotection in the CA1 in the light treated 2VO group?

A7: We thank the reviewer for raising this important question. Yes, indeed blocking RGS12 will be expected to reduce neuroprotection to CA1 in light-treated 2VO group. RGS12 is an important regulator of the N-type calcium channels. There is strong evidence in the literature demonstrating that blockers to N-VGCC are neuroprotective against ischemia (Pringle AK et al., *Stroke*, 1996; Yamamoto T and Takahara A, *Curr Top Med Chem*, 2009; Kamp MA et al., *Stroke Res & Treat*, 2013). Therefore, this interesting question deserves to be fully addressed in a separate paper. Especially we have now gathered a large set of proteomics data indicating several associated protective genes that are activated in this paradigm. We are in the process of putting together concrete experimental data to demonstrate RGS12's role in 2VO neuroprotection in a separate paper. The current study provided a detailed analysis of RGS12's role in modulating neurotransmitter release and presynaptic plasticity. We have discussed this in the revised Discussion on pp. 19 last 4 lines.

Q8. Figure S7, the authors should list the top 5 or 10 up- and down-regulated proteins. Simply describing the numbers is hard to get a sense about the extent of changes upon each treatment.

A8: We agree with Reviewer #1 and have provided the full list of the genes with altered expressions in the revised Fig S7d.

Finally, we would like to thank Reviewer #1 for spending the valuable time to review our manuscript. These constructive and helpful suggestions are much appreciated. The new data strengthened our conclusions and significantly improved the quality of this manuscript.

REVIEWER #2:

The major claims of this paper are that 40 Hz light stimulation induces gamma rhythms in the hippocampus, improves hippocampal-dependent memory, improves synaptic plasticity, and is neuroprotective in a mouse model of stroke. The paper includes a variety of different physiological, behavioral, and anatomical methods, but many of the tests were not done rigorously. The conclusions of the paper are not convincingly supported by the results presented. The hypotheses and rationale for performing many of the experiments are also often unclear. Specific comments are detailed below:

A: We sincerely thank Reviewer #2 for the time spent reviewing our manuscript. The constructive comments and suggestions are much appreciated. We have carefully discussed all the points raised by Reviewer #2 and added new experimental data to address questions raised by this reviewer in a point-by-point manner. In particular, the hypothesis and rationale for performing some of the related experiments are now added in the revised manuscript. Additional controls are also provided to increase the rigor of the data. The newly added results lent further supports to our original conclusions and we believe that the revised manuscript is much improved.

We would like to emphasize that the main focus of the current study was to demonstrate neuroprotective effects conferred by the light flicker at 30, 40 and 50 Hz frequencies. A large amount of data was also devoted to the understanding of RGS12-mediated synaptic functions in response to light treatment during cerebral ischemia. Both of these findings are novel. However, the demonstration of hippocampal oscillatory response is necessary, which confirmed previous publications and also served as a proof-of-concept for the current studies.

Major Points:

Q1. The authors do not convincingly demonstrate that the 40 Hz LED stimulation generates 40 Hz oscillations in the hippocampus. Specific points related to this general point are listed below.

a) In Figure 1C, it appears as though $1/f$ noise differs between normal light and 40 Hz LED groups. Also, the 40 Hz LED stimulation is associated with a sharp peak at 40 Hz, which is not physiological and is instead likely to be a stimulation artifact.

A1a: We thank the reviewer for raising these questions and apologize for the lack of clarity. Measurements of brain activity, such as the local field potentials (LFPs), display $\sim 1/f$ frequency scaling in their power spectra, especially at low frequencies. To confirm the similarity between normal light (Occluded LED in the revised Fig 1) and 40 Hz light treatment groups, we used the Pearson

product-moment correlation coefficient analysis to compare LFP curves between groups. There are high degrees of correlations in LFP curves between random LED and 40 Hz LED groups ($r = 0.98 \pm 0.007$, $n = 6$), between Occluded LED and Random LED groups ($r = 0.99 \pm 0.012$, $n = 6$), and between Occluded LED and 40 Hz LED groups ($r = 0.97 \pm 0.008$, $n = 6$). We have reanalyzed the data and added new plots in the revised Fig 1b-g and Fig S1j-m. These analyses further supported our original conclusion.

40 Hz light flicker entrainment with low gamma oscillation has been demonstrated and published recently by two groups in several high impact papers (Adaikkan et al., *Neuron*, 2019; Martorell et al., *Cell*, 2019; Seo et al., *Nature*, 2016; Etter G et al., *Nat Comm*, 2019, all cited). Spiking entrainment and phase-locking results are strong evidence in support of 40 Hz light entrainment with low gamma as shown below by Prof Tsai L.-H.'s group from MIT:

[copied from Adaikkan C, *Neuron*, 2019, 102(5): 929-943. e8. Fig. S2 A-C]

We used LFP to confirm proof-of-concept the entrainment phenomenon in the hippocampus. In the revised Fig S1j-m, we also provided additional data to demonstrate the occurrence of hippocampal entrainment with 10, 30, 40, 50 Hz light flickers. The arrhythmic frequency light and mice occluded 40 Hz light did not show entrainment (Fig 1b-d). The LFP recordings in CA1 showed a clear increase in the power spectral density at the corresponding frequencies. The newly added results lent further support to our original conclusion.

b) The recordings shown in Figure 1B are too low resolution to see anything.

A1b: We apologize for the lack of clarity in Fig 1B. In the revised Fig1h, the raw recordings of CA1 LFP of four groups of mice are provided at a much higher resolution. The respective LFP powers were also shown in Fig 1i-k.

c) Why are there 2 peaks in Figure 1B (below)? Could these be theta harmonics? What was observed at lower frequencies?

A1c: We thank the reviewer for raising this important issue. The original Fig 1B was a moving window LFP power spectrogram of CA1 time-locked as a mouse initiating approach (< 0 s) and explore (> 0 s) a novel object. An increased PSD ratio of low gamma power during an exploration/approach behavior was used to represent information retrieval between CA3-CA1. This exact experimental paradigm has been used previously to study CA3-CA1 cognitive information flow (Trimper, J et al., Cell Rep 2017, cited). We used this as a readout to indicate a light flicker's effect on CA1 oscillation. To make it clear to the readers, we measured and compared the power of 40 Hz low gamma and found an almost 2-fold increase in intensity between exploration vs approach as shown in revised Fig 1n,o,p. In contrast, ratios for theta and high gamma did not change, which supports our original conclusions.

The potential theta harmonics issue is an important one. Because mouse movement speed affects the linearity of theta waves and potentially presents theta harmonics during high speed running (Zhou et al., eNeuro, 2019; Sheremet et al., 2016, J Neurosci, all cited), we selected new raw data matched to three low-speed movement states, i.e. approach, explore and low-speed free walking. The speed/velocity of mouse movement during approach/exploration and freely walking was between 2 cm/s \sim 3 cm/s forward as determined by footfall patterns (Broom L et al., 2017, cited). These states of movements are relevant to 2VO mice since, within the first few days of recovering from 2VO surgery, the mouse does not normally engage in vigorous movement. Importantly, at these low movement velocities, the spectrograms are not supposed to show prominent theta harmonics as reported by Maurer and colleagues (Zhou et al., eNeuro, 2019; Sheremet et al., 2016, all cited).

In the revised Fig 1, we have added these data to show the quantification of the power of theta, low gamma, and high gamma and theta-gamma phase-amplitude coupling (PAC) of the four groups of mice (Fig 1).

We recognized that our LFP data is limited to address the harmonic issue, which is obviously important and deserves future studies by appropriately fitted laboratories. But supported by published data using spiking entrainment and phase-locking which are resistant to harmonics issues (Adaikkan et al., Neuron, 2019; Martorell et al., Cell, 2019; Seo et al., Nature, 2016, all cited), we trust adequate proof-of-concept data have been provided to allow us to focus more on the downstream protective effects of light treatment. The new movement data are now added in the revised MS on pp. 5 and 18. Thank you.

d) Hippocampal rhythms are affected by animals' behavior (e.g., movement speed). There is no demonstration that behavior was the same across groups. For example, it is possible that the effects were observed because the light

stimulation made the mice run faster.

A1d: We thank Reviewer #2 for raising this important issue and apologize for the lack of clarity in the previous MS. Indeed, the locomotion speed and velocity affect hippocampal rhythms (Cantolty and Knight, 2010; Colgin, 2015; Mably et al., 2018; Sheremet et al., 2019; Zhou et al., eNeuro, 2019, all cited).

We have provided detailed explanations above in A1c for selecting three similar movement states and reported the analysis in the revised Fig 1. Additionally, light treatment does not make mice run faster. We performed the open-field test to demonstrate that light treatment did not induce anxiety-like behavior. Furthermore, within the first few days of recovering from 2VO surgery, the mouse does not normally engage in vigorous movement. Therefore, the new data lent further support to our original conclusions.

e) There are no peaks observed in Figure 1E, just broadband power. It is unclear what is actually being measured here when power is taken from the 30-50 Hz band. Were any differences observed at lower frequencies?

A1e: We apologize for the lack of clarity in explaining the original Figure 1E. The original Fig 1E was a moving window LFP power spectrogram of CA1 time-locked as a mouse initiating approach (< 0 s) and explore (> 0 s) a novel object. The total spectral power density at the indicated frequencies was calculated. The ratio of power spectral density of theta, low gamma, 40 Hz and high gamma between exploration and approach events were compared among the four groups of mice

An increased ratio of low gamma power during an exploration/approach behavior was used to represent information retrieval between CA3-CA1. This exact experimental paradigm of moving window spectrograms has been used previously to demonstrate CA3-CA1 cognitive information flow (Trimper, J et al., Cell Rep 2017, cited). We used this as a readout to indicate the effect of a light flicker on CA1 oscillation. Change of this ratio also indicates cerebral ischemia-induced changes in cognitive function.

As requested by Reviewer #2, in the revised Fig 1, we added the data for theta frequency, redrawn the exploration/approach PSD spectrograms, and provided new calculation plots for $PSD_{\text{exploration}}/PSD_{\text{approach}}$ ratios of theta, low gamma, high gamma and specific 40 Hz low gamma to indicate changes amongst the four experimental groups. Indeed, only the total low gamma PSD and the 40 Hz PSD increased by about 10% to 200%, respectively, during exploration. Theta and high gamma did not change. Importantly, 40 Hz light flicker restored 2VO-induced deficits in low gamma. Again, these results support our original conclusions.

f) The animals' movement speed likely differed during novel object exploration, but the authors did not appear to control for differences in movement speed.

A1f: We thank the reviewer for raising this important question. In the same vein as described above in A1c and A1d, we selected new raw data matched to three low-speed movement states, i.e. approach, exploration and low-speed free walking. The speed/velocity of mouse movement during approach/exploration and freely walking was between 2 cm/s ~ 3 cm/s forward as determined by footfall patterns (Broom L et al., 2017, cited). These states of movements are relevant to 2VO mice since, within the first few days after 2VO surgery, the mouse does not normally engage in vigorous movement. We considered that an exploration event had occurred once the mouse approached and nose sniffed the novel object. Therefore, the approach and exploration are relatively low-speed events. It is important in the future to demonstrate the effect of a mouse running at an increasing speed using a fitted running ball system, which my laboratory currently does not have.

2. Another major problem with the paper is that the authors conclude that 40 Hz stimulation produces the effects that they observe, but, at least for most of the figures, there were no controls performed using other types of sensory stimulation. So, it is unclear whether any type of sensory stimulation, not just 40 Hz light flickering, would produce similar effects as those reported in Figures 4-7.

A2: We thank Reviewer #2 for these important comments. Other sensory stimulations are indeed necessary controls for 40 Hz light flickering. To this end, stimulations with 10 Hz, 80 Hz, and arrhythmic frequency lights were added in the revised Fig S4 serving as controls to demonstrate the lack of rescue of LTP_{NMDAR}, LTP_{L-VGCC}, PPR, and neurotransmitter release. These frequencies did not rescue 2VO-induced changes, therefore serving as negative controls to support our conclusions.

3. Does this light stimulation produce temperature differences that could affect recovery from stroke? One could imagine that those LEDs could produce some heat.

A3: The reviewer's concern is justified and also an important one. Temperature indeed affects stroke outcomes. We routinely used a rectal probe to monitor every mouse during 2VO ischemia and during LED light treatment. There were no visible body temperature changes. Further, the body temperature is maintained at 37 °C with a heating blanket and a fan as described in the revised Methods section. The light flicker treatment was performed in an open-top cage in a well-ventilated room within a temperature-controlled animal house. These

measures minimized possible ambient temperature fluctuation and maintained a consistent body temperature during the treatment.

4. There is no interaction effect reported to support the authors' claim that slow gamma is the hippocampal rhythm that is selectively affected in 2VO mice.

A4: We thank the reviewer for this comment. We have now analyzed the theta-gamma phase-amplitude coupling (PAC) and compared it amongst the four experimental groups as shown in the revised manuscript Fig 1I, m. Several published studies directly support the view that hippocampal slow gamma is selectively affected by 2VO. For example, an acute and long-lasting decrease in the low gamma, but not theta or high gamma, activity has been observed in an anesthetized mouse unilateral hippocampal ischemia (Barth and Mody, *J Neurosci*, 2011, cited). In the anesthetized rat 2VO model, a reduction in hippocampal LFPs and cross-frequency coupling between theta and gamma occurred, which was associated with impaired LTP (Xu et al., *Front. Comput. Neurosci.*, 2013, cited). Our current work represents the first attempt of using a concise and free behaving mouse to investigate the roles of hippocampal low gamma oscillations after 2VO.

Our findings are also indirectly supported by the literature. The fact that low gamma coupling to theta, but not high gamma, occurred in Alzheimer's cognitive deficits (Etter et al., *Nat Comm*, 2019, cited), and that the occurrence of low gamma deficits in human stroke patients also affects cognitive functions (Rabiller et al., 2015, cited) provided indirect support of our work on mouse 2VO. We have added these arguments in the revised Introduction and Discussion.

5. Figure 3B-C does not seem to control for potential differences in running speeds across groups. A similar point applies to Figure 3E. Could this again be due to differences in movement speed?

A5: We agree with Reviewer #2 that controlling speed is important across groups. In the original Fig 3B-C, OFT was performed. The total distance and the distance in the center were measured within a 10 min time window to derive at the average speed. Therefore, reduced speed occurred in the 2VO group. In these experiments, mice were not implanted with electrode and one should not be confused with experiments described in Fig 1. As described in "Gould T.D., Dao D.T., Kovacsics C.E. (2009) The Open Field Test. In: Gould T. (eds) *Mood and Anxiety Related Phenotypes in Mice*. *Neuromethods*, vol 42. Humana Press, Totowa, NJ", the OFT is to determine the spontaneous exploration and anxiety behaviors of mouse.

The original Fig 3E described MWM test. The swimming speed was added in the

revised Fig 3g showing no differences amongst the four groups. Data on different pathlength across the four groups of mice are also added to the revised Fig 3f.

6. Regarding statistical analyses, why was a repeated measures analysis not performed for the LTP measurements?

A6: A repeated measure analysis was unfortunately not appropriate for the LTP measurements since each brain slice is independent. Mouse was killed at 3, 7 and 14 d for LTP experiments.

7. The authors make statements that are not supported up by the results, even if the results were convincingly supported (e.g., "Enhancing CA1 low gamma power using 40 Hz light alleviated 2VO-induced cognitive deficits through the strengthening of CA3-CA1 synapses").

A7: We apologize to the reviewer for the confusion and lack of clarity in statements made. This may be because of the improper usage of the English language. We have extensively revised the text and toned down the inappropriate statements to make sure that statements are supported by facts and experimental data. In the revised MS, the particular offensive statement mentioned by Reviewer #2 has been deleted.

8. The explanation for the different types of light stimulation is unclear. Were the different stimulation patterns (e.g., 10 Hz, 30 Hz, etc.) delivered in different mice?

A8: Thank you for the question. Each mouse received light flickers are at 10 Hz, 30 Hz, 40 Hz, 50 Hz, 80 Hz and an arrhythmic frequency with a 2 h break in between while LFP recordings were performed. At least 5 mice in each experimental group were used for comparison and statistical analysis. We added the descriptions in the revised Methods section on pp27.

9. It is unclear exactly how the novel object exploration fit in. Were all mice subjected to this paradigm? Or only those mice implanted with electrodes? On which days exactly did this occur?

A9: We apologize for the lack of clarity in describing these experimental paradigms. Seven days prior to 2VO ischemia, the electrode was implanted into CA1. After the 2VO surgery, mice received immediate (2h after 2VO) light treatment twice per day for up to 14 d. LFPs were recorded during exploration/approach at 3d, 7d, and 14d after 2VO to determine hippocampal oscillations with cognitive function. Because electrode implantation plus 2VO surgery are really invasive procedures, LFP recordings became incompatible

with several behavioral tests, such as the Morris water maze test. Therefore, only a subset of the mice with implants was subjected to a novel object exploration test as described by Trimpr JB et al., (Cell Rep, 2017, cited). Mice shown in Fig 3 did not receive electrode implantation, but only 2VO with light treatments, to determine long-term behavioral benefits.

Minor Points:

1. The rationale for looking at NMDA-R-independent LTP is unclear, considering that the author's test memory in the Morris water and this type of memory has been shown to require NMDA receptors (a Richard Morris Nature paper from the 1980's).

A: We agree with Reviewer #2 that the rationale for NMDAR-independent LTP needs better justification. We have shown that light treatment rescued the significant reduction in NMDAR-dependent LTPs caused by 2VO. However, NMDARs are not always required for the induction of LTP. In the presence of NMDAR antagonist APV at 50 mM, L-VGCC-dependent LTP can be induced. Both types of LTPs played an important role in cerebral ischemia-induced neuronal death and cognitive impairments (Li et al., J Neurosci, 2007, cited). In particular, L-VGCC activity transmits calcium-regulated survival signaling to the nucleus that supports learning and memory (Wang et al., J Neurosci 2016, cited). Inhibiting L-VGCC and its LTP causes delayed death of CA1 neurons in 2VO (Li et al., J Neurosci, 2007, cited). Therefore, the responses of both types of LTPs to flicker light treatment were examined. We have added justifications for conducting L-VGCC LTP in the revised MS on pp. 11,

2. The rationale for the open field test was unclear. Why would the 2VO manipulation be expected to affect anxiety or behavior on this test?

A: The reviewer is correct that 2VO manipulation does not cause elevated anxiety behavior *per se*. OFT was conducted to demonstrate whether light treatment causes any anxiety-like behaviors. The result of OFT is an important indication that the light treatment is benign enough not to make the test mouse behave very differently, such as run faster, which could alter the dynamics of CA1 oscillations.

Again, we would like to thank Reviewer #2 for providing the critical and constructive reviews which made a significant improvement to the quality of this manuscript. The additional experiments and results lent further support to our original conclusions.

Reviewer #3

This is an interesting study which aims to determine if entrainment of hippocampal gamma could alleviate stroke-induced cognitive dysfunction. The authors present the case that post-stroke motor and cognitive dysfunctions are associated with the perturbation of specific brain oscillation frequencies and, importantly, such oscillations have the potential to be non-invasively and externally modulated. This is a comprehensive study which has the potential to provide novel and interesting data to the field. However, some methodological concerns need addressing;

The authors have performed two-vessel occlusion of the common carotid arteries which produces a selective damage of the CA1 pyramidal neurones – the authors need to be clear what (disease) they are modelling – 2VO offers a global and transient ischemia that more clinically represents cardiac arrest rather than the regional focal ischemia seen following cerebral stroke which is achieved via other experimental methods. I think it is rather an over extrapolation to suggest that results obtained from this study may therefore have clinical relevance as a therapeutic for stroke prior to such work being conducted in a regionally focal model of stroke.

A1: We thank Reviewer #3 for the encouraging comments and we really appreciate the helpful suggestions. Specifically, we agree with Reviewer #3 that the current study was to use CA3-CA1 as a system to examine the effect of light treatment. Therefore, the offensive statements about clinical relevance as therapeutics for stroke have been removed to tone down the impact of the conclusions. The MS has been revised accordingly.

The justification of the inclusion of sham animals is unclear given the statistical comparison is 2VO vs 2VO+LED.

A2: We apologize for the lack of clarity in explaining why the Sham and Sham+LED groups were used. The reviewer is correct that most of the comparisons were made between 2VO and 2VO+LED groups. The use of the Sham group is, however, necessary as a control for comparison with the 2VO-induced specific brain damage and LED light on the mouse. Moreover, Sham mice are necessary as a control to determine the light entrainment. Statistics were performed in several figures to demonstrate significant differences between Sham vs 2VO, and Sham vs Sham+LED groups. Therefore, Sham and Sham+LED animals are required controls.

It's unclear why only 5 animals were used to determine ischemia/reperfusion-induced changes in rCBF? Were the same 5 animals used to determine the changes in MABP?

A3: We apologize for the lack of clarity in describing measurements in blood flows. The systemic main arterial blood pressure (MABP) of every mouse underwent 2VO and sham surgeries were routinely measured to maintain hypotension during surgery at 35-40 mmHg (n=20). The rCBF of the hippocampus was measured on 5 mice per group (2VO and 2VO+LED) to determine whether light treatments affected rCBF. The detailed descriptions were added to the revised Methods section is on pp. 25-26, 34-35 and Results section on pp. 9.

In vivo experiments should be reported according to the Arrive guidelines and therefore include details of; total animal used, inclusion/exclusion criteria, how animal numbers were determined, if any, and for what reasons, animals were excluded.

A6: We thank Reviewer #3 for this important suggestion. Adhering to the ARRIVE guideline for animal studies is important. We have now added the required information related to ARRIVE guidelines in the revised Methods sections to provide the required information concerning the total animals used, inclusion/exclusion criteria, how animal numbers were determined, and for what reasons animals were excluded. These revisions are listed on pp. 23 under the sub-sections of "Ethical approval and animal study design", and "Mouse two-vessel occlusion (2VO) ischemia model".

In some instances (e.g. Morris water maze) n values (4) appear low – how do the authors know the appropriate number of animals were used?

A7: For the Morris water maze results shown in Fig 3, the numbers of mice per treatment group were n = 10, which should not be confused with the 4 groups of experimental mice studied.

Presentation of results is a little confusing as does not match order of experimental detail provided in methods. For example, I would have expected reporting of rCBF and MABP data first. Also it would be useful for the authors to show confirmation of CA1-selective damage following the 2-VO. The authors need to show that the two x 2VO experimental groups received the same amount of damage (i.e. change in rCBF) and that the only difference between the groups was the presence, or absence, of LED.

A8: We thank Reviewer #3 for raising these important issues. In the revised MS, measurements of rCBF were described in more detail in the Result section on pp. 9. We also provided additional data using laser speckle imager to monitor cortical CBF and rCBF in the visual cortex area changes at 3 d, 7 d, and 14 d post 2VO in response to light treatment. Furthermore, the blood vessel

densities in the hippocampus and visual cortex were also shown using FITC-Lectin staining. These results are presented in the revised MS on pp. 9 and FigS1 and the Methods section on pp. 25, 34. These results addressed Reviewer #3's question that the light treatment did not cause a difference in regional blood flow.

It is indeed an important issue that selective CA1 damage is shown in the current model. Questions along the same vein were also raised by Reviewer #1 (Reviewer #1 A2 and A3). To this end, we provided a detailed characterization of the 2VO model in the revision on pp. 8-9 and Fig. S2a-c. Supported by the published papers (Kristian & Hu, 2013, cited; Onken et al., 2012, cited), there is an occlusion time-dependent increase in damages in brain areas in the 2VO model. Moreover, occlusion for more than 5 min began to increase the death rate of mice and also the death of neurons in the DG area. Therefore, careful control of the systemic hypotension (MABP performed in the current study) and monitor of the hippocampal regional blood flow (hrCBF) proved to be essential to minimize ischemic damages to other brain areas. We added additional images to show no FJB positive cells in other brain areas following 2VO (Fig S2a). We also provided quantifications to demonstrate no significant loss of parvalbumin-positive interneurons in the hippocampus following 2VO (Fig S2b,c). Together, these new data lent further support to our original conclusions.

Finally, we thank all three reviewers for their valuable constructive suggestions. We have provided the required additional data to strengthen our conclusions, which definitely improved the quality of this manuscript.

Reviewer #1 (Remarks to the Author):

Zheng et al. presented a revised version of their ms describing the neuroprotective effect of 40 Hz flickering light on CA1 neurons, synaptic plasticity of Hp and in vivo cognitive function in a 2VO+hypotension global ischemia model. The authors have done a good job in responding to prior critiques with either new data or useful discussions. I am largely satisfied with the responses to my comments. I have only minor requests or suggestions for the authors to consider below:

1. It would be more convincing to show Fluorojade staining of frontal cortex, striatum and thalamus nuclei of mice with 2VO to fully characterize this model.
2. I suggest to cite ref #46 as the reference for the 2VO model on page 23, since ref #19 was a rat study.
3. For the hippocampal LFP-novel object exploration study, direct evidence in moving velocity would be helpful to strengthen the data.

Reviewer #2 (Remarks to the Author):

I appreciate the authors' efforts to revise the manuscript, and many aspects have improved. However, my concerns regarding the rigor of the statistical analyses were not really addressed or were misunderstood. I repeat my concerns again below and try to clarify so that the authors will understand what I mean:

1. Statistics should be clearly described in the text, and statistics should be provided with degrees of freedom and exact p-values, not just p-values < 0.05 or whatever.
2. Main effects in ANOVAs should be reported to justify use of post-hoc tests.
3. When comparing effects of treatments across groups, all groups should be included in the same analysis, with a significant interaction effect shown to back up the claim (for example, in Figure 1, authors conclude that different rhythm types are differentially affected by the different types of treatment, but do not report a significant interaction effect to back up this claim). It is inappropriate to conclude that differential effects of treatments are observed across groups simply because p-values are significant for one group and not another (See <https://www.nature.com/articles/nn.2886>). For example, in figure 1, it is inappropriate to conclude that only low gamma is affected simply because p-values are significant for low gamma group and not for the theta and high gamma groups.
4. Repeated measures tests should be used when multiple measurements are taken from the same slices or animals across time, when different rhythm types are analyzed within the same recordings, etc.

Reviewer #3 (Remarks to the Author):

Manuscript has been updated with specific reference to reviewer comments. In particular the clarity (in particular of methods and results) has improved along with reducing the over interpretation of the impact/translational relevance.

Claire Gibson

REVIEWER COMMENTS

Reviewer #1 (Remarks to the Author):

Zheng et al. presented a revised version of their ms describing the neuroprotective effect of 40 Hz flickering light on CA1 neurons, synaptic plasticity of Hp and in vivo cognitive function in a 2VO+hypotension global ischemia model. The authors have done a good job in responding to prior critiques with either new data or useful discussions. I am largely satisfied with the responses to my comments. I have only minor requests or suggestions for the authors to consider below:

A: We thank Reviewer #1 for the overall very helpful and constructive comments. We have provided the requested information in the revised MS, exactly as suggested. Thank you.

1. It would be more convincing to show Fluorojade staining of frontal cortex, striatum and thalamus nuclei of mice with 2VO to fully characterize this model.

A1: We agree with Reviewer #1 and have provided the FJB staining for the frontal cortex, striatum, and thalamus nuclei in the revised Supplementary Fig. 2a. We did not see apparent cell death in these regions of the brain, supporting our original conclusions.

2. I suggest to cite ref #46 as the reference for the 2VO model on page 23, since ref #19 was a rat study.

A2: Reviewer #1 is correct. It is corrected!

3. For the hippocampal LFP-novel object exploration study, direct evidence in moving velocity would be helpful to strengthen the data.

A3: Thank you for the suggestion. We have provided the measurements of the moving velocity of three locomotive states shown in the revised Supplementary Fig. S1n, which demonstrated that during LFP-novel object approach, exploration, and free slow moving, the mouse moving velocity was at 2 -3 cm/s.

Reviewer #2 (Remarks to the Author):

I appreciate the authors' efforts to revise the manuscript, and many aspects have improved. However, my concerns regarding the rigor of the statistical analyses were not really addressed or were misunderstood. I repeat my concerns again below and try to clarify so that the authors will understand what I mean:

1. Statistics should be clearly described in the text, and statistics should be provided with degrees of freedom and exact p-values, not just p-values < 0.05 or whatever.

A1: We sincerely thank Reviewer #2 for raising this critical issue and apologize for the lack of clarity in reporting statistics. We have now carefully examined all statistical analyses and provided the required statistical information (degrees of freedom and exact p-values, not just values < 0.05) in the revised MS, as suggested by Reviewer #2. The revisions are in the Results and Figure Legends sections which are highlighted in red colors.

2. Main effects in ANOVAs should be reported to justify use of post-hoc tests.

A2: We agree with Reviewer #2. Now the main effects of ANOVAs are reported in the revised MS Results. For example, Fig 1 i-m, o-r; Fig 2c-e; Fig 3b-h; Fig 4a-g; Fig 5a,b,d,e; Fig 6e-l; Fig 7d; Fig 8a-l; and those for the Supplementary Figs as indicated in the appropriate figure legends. All revisions and additions are highlighted in red color text. Thank you.

3. When comparing effects of treatments across groups, all groups should be included in the same analysis, with a significant interaction effect shown to back up the claim (for example, in Figure 1, authors conclude that different rhythm types are differentially affected by the different types of treatment, but do not report a significant interaction effect to back up this claim). It is inappropriate to conclude that differential effects of treatments are observed across groups simply because p-values are significant for one group and not another (See <https://www.nature.com/articles/nn.2886>). For example, in figure 1, it is inappropriate to conclude that only low gamma is affected simply because p-values are significant for low gamma group and not for the theta and high gamma groups.

A3: We sincerely thank the Reviwer#2 for pointing out these potential errors in statistical analyses. We have read the suggested article carefully and fully understood the need that when making a comparison between two effects, one should report the statistical significance of their difference rather than the difference between their significance levels. In the revised manuscript, we have now provided analyses of the interaction effects and simple main effects, if there was a statistically significant interaction effect, to back up our claims. For example, Fig 1i-m, o-r; and all two-way ANOVA analyses.

4. Repeated measures tests should be used when multiple measurements are taken from the same slices or animals across time, when different rhythm types are analyzed within the same recordings, etc.

A4: We agree with Reviewer #2 and apologize for not performing the repeated measures in some of the studies. In the revised MS, repeated-measures ANOVAs (RM-2ANOVA) were performed for studies involving time course treatments on the same subject such as the LTP studies, behavioral studies, and animals across the time when different rhythm types are analyzed within the same recordings. For example, Fig 1o-r; Fig 3b-h; Fig 4a,b, d-g; Fig 5a-b,d-e; Fig 8a-l. Supplementary Fig S4 a, b, d, e, h, j, l. These statistical analyses strengthened our conclusions.

Again, we wish to thank Reviewer #2 for the constructive and very helpful comments which improved our MS tremendously.

Reviewer #3 (Remarks to the Author):

Manuscript has been updated with specific reference to reviewer comments. In particular the clarity (in particular of methods and results) has improved along with reducing the over interpretation of the impact/translational relevance.

Claire Gibson

A: Thank you.